

# Transporter gene family evolution in ectomycorrhizal fungi in relation to mineral weathering capabilities

Katharine A. King, Petra Fransson, Roger D. Finlay, Marisol Sánchez-García

Department of Forest Mycology and Plant Pathology, Swedish University of Agricultural Sciences, Uppsala, 750 50, Sweden

*Correspondence to*: Katharine A. King (katharine.king@slu.se) and Marisol Sánchez-García (marisol.sanchez.garcia@slu.se)

**Abstract.** The role of ectomycorrhizal (ECM) fungi in biological mineral weathering is increasingly recognised, although the quantitative significance of microbially mediated mineral dissolution for plant growth is debated. Species within the ECM genus *Suillus* are found to preferentially inhabit mineral soils and are frequently reported to possess mineral weathering capabilities. Though studies growing ECM fungi with minerals have shown heightened mycelial nutrient content compared to growth without minerals, the mechanistic understanding of nutrient mobilisation and uptake associated with

weathering remains largely unknown. We examined copy numbers of 173 transporter gene families present in 108 Agaricomycetes species, and analysed evolutionary expansions and contractions of base cation transporter gene families across the phylogeny. We also quantified mycelial base cation uptake by ECM species in the genera *Suillus* and *Piloderma*, and two saprotrophic fungi, when grown in pure culture with and without minerals. We hypothesised that 1) greater base cation uptake is dependent on evolutionary expansions in copy-numbers of base cation transporter genes, 2) mineral

weathering results in base cation uptake by *Suillus* growing in pure culture with mineral additions, 3) base cation uptake by *Suillus* growing in pure culture with mineral additions will be greater compared to other fungal species, and 4) base cation uptake will correlate with base cation transporter gene family copy numbers. We showed that 25 transporter gene families are significantly expanding and contracting in the genus *Suillus*, 10 of which correspond to base cation transporter families and two of which are accessories to base cation transport, highlighting the importance of base cation uptake and transport in

the life strategy of *Suillus* species. Additionally, there are significant expansions in the Fungal Mating-type Pheromone Receptor (MAT-PR) Family in *Suillus* species, suggesting that these species are adapting to their environment. For all elements there were examples of higher mycelial concentrations after growth in pure culture with mineral additions as compared to the nutrient limited treatment for *Suillus* species but also for some *Piloderma* species, confirming that mineral weathering resulted in base cation uptake in pure culture. For 40% of the significantly expanding and/or contracting base

cation gene transporters families, copy numbers were significantly correlated with uptake of mineral elements, and most of



the significant correlations were positive. This suggested that members of the genus *Suillus* are rapidly evolving, and that the expansions and/or contractions in these transporter families are likely to be related to the requirement of base cation uptake, underpinned by the finding that mycelial uptake of elements in the presence of minerals often increased with base cation transporter family copy numbers. Further work into other levels of regulation, e.g. through transcriptional regulation of

transporter proteins, would be useful to gain a deeper understanding of base cation uptake regulation and its role in mineral weathering by *Suillus* species, and ECM fungi in general.

## 1 Introduction

In boreal forests biological mineral weathering is driven by plant-associated ectomycorrhizal (ECM) fungi, which form the

dominant type of symbiosis, primarily with trees of the family Pinaceae (Smith and Read, 2008). The ECM symbiosis plays key roles in plant nutrient acquisition, carbon (C) partitioning and biogeochemical cycles by exchanging nutrients mobilised by ECM fungi from the soil with photosynthetically derived C from the host plant (Clemmensen et al., 2013; Genre et al., 2020; Van Der Heijden et al., 2015). Essential mineral nutrients, or base cations, and phosphorus (P) and nitrogen (N) are primarily mobilised by mineral weathering and decomposition of organic matter, respectively (Finlay et al., 2020; Hoffland

et al., 2004; Tunlid et al., 2022). Boreal forest soils are stratified podzols, with an overlying organic (O), a middle mineral eluvial (E), and a typically thicker underlying mineral illuvial (B) soil horizon, which form due to low temperatures and pH, recalcitrant plant litter, slow decomposition rates, and the absence of earthworms resulting in a lack of mixing (van Breemen et al., 2000; Lundström et al., 2000). The O horizon is the main source of N (Hobbie and Horton, 2007) and holds the greatest biological activity, with the highest density of short roots, and ECM mycelial biomass in similar quantities to that of

fine roots (Wallander et al., 2001). ECM fungi dominate the soil fungal communities both in the fermentation and humus layers of the O horizon, and in the E and B horizons (Bödeker et al., 2016; Lindahl et al., 2007; Marupakula et al., 2021). Mobilisation of base cations via mineral weathering by ECM fungi occurs mainly in the E and B horizons (van Breemen et al., 2000; Jongmans et al., 1997; Mahmood et al., 2024) and even though fine root density is lower compared to the O horizon, due to the large volume the mineral soil contains up to two thirds of ECM root tips, up to half of all ECM taxa

across a boreal forest profile, and many ECM taxa which are not found elsewhere (Rosling et al., 2003). Although decomposition of organic matter to acquire N and mineral weathering to acquire base cations predominantly occur separately in discrete soil horizons, these processes are integrated. Mahmood et al., (2024) demonstrated that weathering in the mineral soil was driven by sinks generated by ECM mobilisation of N from the O horizon, and that changes in organic matter availability in the O horizon influenced fungal C allocation to deeper mineral horizons, as well as the abundance of different

functional fungal guilds. Weathering of base cations in the B horizon can be upregulated in response to demand induced by improved N supply from the O horizon to both the plant and the ECM fungus, but when organic matter is depleted, both N acquisition and mineral weathering rates are reduced. Mahmood *et al.*, (2024) also showed that mobilisation and uptake of



magnesium (Mg), which acts as an activator for many enzymes and plays a key role in photosynthesis (Black et al., 2007; Bose et al., 2011), occurred primarily in the deeper mineral horizon and was driven by carbon allocation to ECM mycelium.

Intense forestry practices and the increasing demand for bioenergy result in a reduction of organic matter supplied to the O horizon and the return of nutrients, like N, P and base cations, to the soil (Akselsson et al., 2007, 2019; Klaminder et al., 2011; Moldan et al., 2017). These practices are suggested to be unsustainable for long-term plant nutrition, to alter fungal communities and to negatively impact mineral weathering rates and base cation supply (Finlay et al., 2020; Joki-Heiskala et al., 2003; Rähn et al., 2023). Therefore, a better understanding of mineral weathering, base cation uptake and the key role

ECM fungi play in these processes is essential in the development of more sustainable forestry practices.

Fungi can mobilise base cations from mineral soils by physical force and extrusion of low molecular weight organic acids (LMWOAs), free radicals, protons and siderophores, which consort together to weather minerals (Finlay et al., 2020; Fomina et al., 2010; Schmalenberger et al., 2015). In particular, LMWOAs are known to play a significant role and mineral

weathering by ECM fungi has been demonstrated in a number of studies (Calvaruso et al., 2013; Drever and Stillings, 1997; Paris et al., 1996; Schmalenberger et al., 2015; van Scholl et al., 2006; Wallander, 2000). Members of the genus *Suillus*, many of which grow well in culture and are commonly used in experiments, have been frequently reported to perform mineral weathering and are found primarily in the B horizon in boreal forests (Lofgren et al., 2024; Mahmood et al., 2024; Marupakula et al., 2021). In the presence of minerals, several *Suillus* species have been observed to increase production and

exudation of LMWOAs, a pattern not seen in the absence of minerals (Adeleke et al., 2012; Olsson and Wallander, 1998; Wallander and Wickman, 1999). *Suillus* species have also been shown to take up base cations, for example potassium ($K^+$), magnesium ($Mg^{2+}$) and iron ($Fe^{2+}$) was taken up by *S. tomentosus* growing in liquid culture with the mineral biotite (Balogh-Brunstad et al., 2008a), and *S. variegatus* was shown to take up $Mg^{2+}$ in pure culture with granite rock (Fahad et al., 2016). Mahmood *et al.*, (2024) found that $Mg^{2+}$ was taken up from B horizon soil solution in reconstructed podzol microcosms

containing *Pinus sylvestris* seedlings growing in natural forest soils in which *S. bovinus* was the most abundant taxon. Other mineral weathering ECM species include *Paxillus involutus*, which when grown in pure culture has been shown to weather biotite (Bonneville et al., 2009), chlorite (Gazze et al., 2012), muscovite (Van Scholl et al., 2006), and phlogopite (Paris et al., 1995, 1996), and *Pisolithus tinctorius* which has been shown to increase mineral weathering rates when in symbiosis with *Pinus resinosa* seedlings (Balogh-Brunstad et al., 2008b). These previous studies however, relied on abundance data

and chemical analyses of ECM fungal biomass, plant biomass, or growth medium of experimental systems, and did not focus on the mechanism or regulation of base cation uptake.

Regulation of base cation uptake through transporter proteins can occur at the genomic, transcriptional, post-transcriptional, translational and post-translational level (Mukhopadhyay et al., 2024). One mechanism of regulating transporter protein



expression is copy number variation of a protein encoding gene. This has been shown to drive genome evolution and

environmental adaptation (Steenwyk and Rokas, 2017; Tralamazza et al., 2024). Wapinski *et al.* (2007) found that genes

related to growth and maintenance show little duplication or loss in copy number, but genes related to stress and

environment interactions show much greater fluctuations in copy number. Genes encoding transporter proteins belong to the

latter group, thus, it is reasonable to expect lifestyle and niche dependent fluctuations in copy number as a form of adaptation

and regulation of transport. Fungal comparative genomic studies have highlighted the importance of gene family copy

number in the emergence of the ECM lifestyle from saprotrophic ancestors (Kohler et al., 2015; Miyauchi et al., 2020). Gene

family copy numbers of plant cell wall degrading enzymes, important in fungal-mediated decomposition of plant material,

have been observed to be markedly reduced in ECM fungi compared to their saprotrophic ancestors, and gene copy numbers

of symbiosis-induced secreted proteins have been shown to expand in ECM fungi (Miyauchi et al., 2020). Studies focusing

on regulation of base cation transporter genes are scarce, however, upregulation of $K^+$ transporter genes during mineral

weathering has been identified in *Hebeloma cylindrosporum* (Corratgé et al., 2007; Garcia et al., 2014), *Amanita pantherina*

(Sun et al., 2019) and *Paxillus involutus* (Pinzari et al., 2022). Whilst there are some studies into the regulation of base

cation transporter genes, there are currently no studies investigating the evolution of base cation transporter genes involved

in ECM-mediated mineral weathering.


Here, we investigated the evolution of base cation transporter gene families in the Agaricomycotina, with a particular focus

on the genus *Suillus*. We hypothesised that 1) greater base cation uptake is dependent on evolutionary expansions in copy-

numbers of base cation transporter genes, 2) mineral weathering results in base cation uptake by *Suillus* growing in pure

culture with mineral additions, 3) base cation uptake by *Suillus* growing in pure culture with mineral additions will be

greater compared to other fungal species, and 4) base cation uptake will correlate with base cation transporter gene family

copy numbers.  Using bioinformatics we explored evolutionary expansions and contractions in base cation transporter gene

families, and using experimental approaches we determined base cation uptake by five species of *Suillus* (*S. bovinus*, *S.*

*granulatus*, *S. grevillei*, *S. luteus* and *S. variegatus*), and a selection of other ECM and saprotrophic fungi that were grown in

pure culture with and without mineral additions. This study aims to shed light on the evolution of base cation transporter

genes and the importance of their copy numbers in relation to mineral weathering and base cation uptake by ECM fungi and

in particular, the genus *Suillus*.

## 2 Materials and methods

### 2.1 Bioinformatics

#### 2.1.1 Agaricomycotina phylogeny

A phylogeny of 108 published Agaricomycotina species was inferred using genome data downloaded from JGI MycoCosm

(https://mycocosm.jgi.doe.gov/mycocosm/species-tree/tree;aY7X8o?organism=agaricomycotina) (Sup. Tab. 1). *Tremella*



*mesenterica* (Tremellomycetes) and *Dacryopinax primogenitus* (Dacrymycetes) were used as outgroups. Orthogroups were identified with OrthoMCL v2.0.9 (Chen et al., 2006) with the settings percentMatchCutoff=50 and evalueExponentCutoff=-5. Single copy orthogroups that were present in at least 50% of the taxa were chosen for the subsequent analyses. Multiple sequence alignment was done with MAFFT v7.407 (Katoh et al., 2005) using the auto settings after which poorly aligned regions were removed with TrimAL v1.4.1 (Capella-Gutiérrez et al., 2009) with a gap threshold setting of -gt 0.1. Individual gene alignments were concatenated with genestitcher v3 (https://github.com/ballesterus/Utensils). Finally, a maximum likelihood phylogeny was reconstructed with IQ-TREE v2. (Nguyen et al., 2015) using the MFP+MERGE arguments to select the best-fit substitution model using ModelFinder (Kalyaanamoorthy et al., 2017) and with 1000 ultrafast bootstraps (Hoang et al., 2018).

**2.1.2 Molecular dating**

The resulting phylogenetic tree was time calibrated with r8s v1.81 (Sanderson, 2003) using the penalised likelihood method, POWELL optimisation and fossil constrained cross validation. Multiple smoothing parameters were tested before selecting a value of 0.01 which gave the best cross validation score. The root of Agaricomycotina was fixed to 436 million years ago (MYA). A suilloid ectomycorrhizal fungi fossil was used to calibrate the Suillaceae node (~50-100 MYA) (LePage et al., 1997) and the fossil *Palaeoagaricites antiquus* was used to calibrate the Agaricales node (~105-210 MYA) (Poinar and Buckley, 2007). The resulting ultrametric tree was then visualised and edited with inkscape (https://inkscape.org).

**2.1.3 Analysis of transporter gene family evolution**

Transporter gene family annotations were downloaded from JGI MycoCosm (https://mycocosm.jgi.doe.gov/mycocosm/annotations/browser/tcdb/summary;FdfSB_?p=agaricomycotina) (Tab. S2), which are categorised to three levels of criteria, following the Transporter Classification Database (TCDB) system (Saier, 2006; Saier et al., 2009, 2014, 2016, 2021). 173 transporter gene families which were present in the analysed taxa were categorised as either being involved in base cation transport or not. Both categories were included in the analysis to facilitate a comparative evaluation of base cation transporter gene evolution. In addition, transporter families which may be rapidly evolving were carefully evaluated, as this together with the presence of a high number of copies of base cation transporter genes, may indicate an increase in mineral weathering capabilities. This data was analysed by principal component analysis (PCA) using CANOCO 5.10 (Microcomputer Power, Ithaca, NY, USA; Braak and Smilauer, 2012) for visualization of relationships between transporter gene family copy numbers and fungal taxa. Additionally, to detect significant differences in copy number among transporter gene families, the influence of taxa and transporter gene family on copy number was tested using a two-way analysis of variance (ANOVA), and copy numbers for each of the transporter gene families were compared using Tukey's HSD in R (packages: AICcmodavg (Mazerolle, 2023), emmeans (Lenth, 2024), multcomp, multcompView (Hothorn et al., 2008)). The same data was used to estimate gene family evolution of transporters across the



phylogeny using CAFE5 v5.1 (Mendes et al., 2020). Due to the birth-death model employed by CAFE5 transporter gene families with zero gene copies at the root were excluded, reducing the analysed transporter gene families from 173 to 114. CAFE5 was unable to run when within-family size variation exceeded 80 copy numbers. Due to this limitation, we ran CAFE5 with a scaled version of the copy number data. Raw gene copy numbers of transporter gene families were scaled to values ranging between 0 and 80 and then rounded up to the nearest whole number in R (packages: tidyverse (Wickham et

al., 2019), scales (Wickham et al., 2024)). This dataset is referred to as the "full" dataset. CAFE5 was also run on a smaller dataset with non-scaled copy numbers to account for any error introduced by the scaling, which included 28, taxa selected as they or their close relatives were present in the pure culture experiment. These taxa were extracted from the original Agaricomycotina tree using R (packages: Geiger (Harmon et al., 2008), ape v5.8 (Paradis and Schliep, 2019)) and will hereby be referred to as the "partial family" dataset. For this dataset, the number of transporter gene families included in

CAFE5 analyses was 126.

Superfamilies of transporter genes which were significantly expanding or contracting in the partial family dataset were further separated into transporter families with four levels of criteria, as these gene superfamilies had high copy numbers and incorporated a broad range of functions making it difficult to determine their precise role. This dataset is hereby referred to

as the "partial subfamily" dataset, with 51 transporter gene families included in CAFE5 analyses (Tab. S3). The partial family and partial subfamily datasets were run in CAFE5 again with the 28 taxa tree. The resulting data was then filtered in two ways to either include only significant expansions and contractions of all transporter gene families, or to include only significant expansions and contractions of base cation transporter gene families and accessories to base cation transporter gene families, using the python script CafeMiner (https://github.com/EdoardoPiombo/CafeMiner/tree/main). Trees were

visualised using CafePlotter (https://github.com/moshi4/CafePlotter).

### 2.1.4 The $Mg^{2+}$ transporter-E (MgtE) family phylogeny

Magnesium is an important base cation which is essential to photosynthesis (Black et al., 2007) and an activator in many enzymes (Bose et al., 2011), and has been previously found to be mobilised and taken up by ECM fungi from mineral soils

(Fahad et al., 2016; Mahmood et al., 2024). As such, we inferred the evolutionary relationships of proteins in the $Mg^{2+}$ transporter-E (MgtE) gene family (TCDB ID 1.A.26). A phylogenetic tree of all MgtE family protein sequences present in the 28 taxa of the partial family dataset was constructed. Protein sequences were extracted from the corresponding taxa proteomes using SAMTOOLS (Danecek et al., 2021) the tree was then reconstructed using the same methods as in section 2.1.1 and visualised in R using the ggtree package (Wang et al., 2020; Yu, 2020; Yu et al., 2017).




**2.2 Base cation uptake in pure culture**

As members of the ECM genus *Suillus* have been reported to possess mineral weathering capabilities the genus was studied in greater depth. Isolates of *Suillus* species, along with the ECM genus *Piloderma* and two saprotrophic fungal species for comparison, were grown in pure culture in axenic systems with and without mineral additions to determine base cation

uptake during mineral weathering.

**2.2.1 Fungal isolates**

A total of 23 fungal isolates, covering 11 species and two isolates identified to genus level, were used in the study (Tab. S4). Members of the ECM genus *Suillus* were isolated in summer and autumn of 2022 from morphologically identified

fruitbodies collected in Central and Southern Sweden. From these a subset of four *S. bovinus*, three *S. granulatus*, one *S. grevillei*, three *S. luteus* and three *S. variegatus* isolates were selected based on geographical spread and growth performance. The genus *Piloderma* was selected as an ECM comparison to *Suillus*, as members of this group are also common in boreal forests and found both in the organic and mineral horizons (Mahmood et al., 2024; Marupakula et al., 2021). Six *Piloderma* isolates, including *P. byssinum*, *P.* aff. *fallax*, *P. olivaceum*, *P. sphaerosporum*, *Piloderma* sp. 1 and

*Piloderma* sp. 2, were obtained from the culture collection maintained at the Department of Forest Mycology and Plant Pathology, SLU. In addition, two saprotrophic brown-rot wood decaying species, *Coniophora puteana* and *Serpula lacrymans* (obtained from Geoffry Daniel at the Department of Forest Biomaterials and Technology, Wood Science, SLU), were included as comparison to the ECM lifestyle. Prior to the experiment, all isolates were grown on Modified Melin-Norkrans (MMN) media (Marx, 1969) for 12 and 5 weeks for ECM and saprotrophic fungi, respectively.


**2.2.2 Experimental conditions**

To measure base cation uptake during mineral weathering, fungal isolates were grown in 9 cm Petri dish closed axenic systems with and without mineral additions, as previously described by (Fahad et al., 2016). Granite and gabbro rock material was selected as mineral sources as they commonly occur in bedrock in Sweden (Holtstam et al., 2004), and are

proposed to have differing weathering rates, with gabbro weathering faster than granite (Bazilevskaya et al., 2013; Fritz, 1988). Granite is a complex of K feldspar, muscovite mica and quartz and gabbro is a complex of sodium (Na)/calcium (Ca) feldspar, amphibole, pyroxene and olivine (Goldich, 1938). These minerals contain essential base cations, such as Ca, iron (Fe), K, Mg and Na, which are required for growth. Granite and gabbro material was provided by Swerock AB, Ängelholm, Sweden, and originated from Stora Almby and Harbo, respectively.  Pulverised rock material was sieved under flowing

deionised water to retain a size fraction of 63-250 mm to maximise surface area. No smaller fraction was included to ensure the material remained in its mineral form and that fungi must weather the sand to access the nutrients. Sand was sonicated



for 10 minutes and sieved again to dislodge and remove any debris adhering to the grains. The sand was then placed in a 1 L

cylinder with small holes at the base and placed under deionised flowing water for 3 days to remove any loosely bound ions.

Two treatments with granite and gabbro, with 30 replicates per isolate, were established. A primary mineral free layer of 20

mL agar media containing $(NH4)_2HPO_4$ 0.22 g/L, thiamine-HCl 100 mg/L, glucose 5 g/L, malt extract 5 g/L and agar 12 g/L

served as a C and N source. A secondary 5 mL layer of the same media with the addition of 20 g/L granite or gabbro,

respectively, was added after the first layer had set. In both cases the pH was adjusted to 5.5 (Fahad et al., 2016).

Two controls with 30 replicates per isolate were also established. These were termed "limited", which provided a minimal

mineral nutrient source, and "rich", which provided a more abundant mineral nutrient source. The limited control had the

same contents as the primary layer described above and was used as a base level comparison against the mineral treatments.

The rich control contained $CaCl_2.2H_2O$ 0.05 g/L, NaCl 0.025 g/L, $MgSO_4.7H_2O$ 0.15 g/L, $(NH4)_2HPO_4$ 0.22 g/L, $KH_2PO_4$

0.5 g/L, $FeCl_3.6H_2O$ 0.012 g/L, thiamine-HCl 100 mg/L, glucose 5 g/L, malt extract 5 g/L and agar 12 g/L, thus providing

all essential nutrients necessary for growth. It provided a comparison of base cation uptake between conditions where

weathering is required to access base cations and where base cations are readily available. Both were set to pH 5.5. Table 1

shows the concentration of Ca, Fe, K, Mg, Na and P present in each of the treatments relative to the limited treatment.

Cellophane was thoroughly rinsed in deionised water and autoclaved, after which it was laid on each plate to prevent

mycelial growth into the agar to aid harvesting, but allow transfer of soluble compounds.

|  | Ca (g/L) | Fe (g/L) | K (g/L) | Mg (g/L) | Na (g/L) | P (g/L) |
|---|---|---|---|---|---|---|
| gabbro | 0,1995 | 0,2770 | 0,0418 | 0,1113 | 0,0141 | 0,0085 |
| granite | 0,0238 | 0,1396 | 0,0115 | 0,0169 | 0,0041 | 0,0049 |
| rich | 0,0136 | 0,0248 | 0,1437 | 0,0244 | 0,0000 | 0,1138 |

Table 1 - Amount of additional base cation and phosphorus (P) nutrients in the gabbro, granite and rich medias compared to

the limited media. Base cations include calcium (Ca), iron (Fe), potassium (K), magnesium (Mg) and sodium (Na).

Concentrations given in g/L.

All isolates were grown in 24 h darkness at 25° C apart from *S. lacrymans* which was grown at 20° C. Due to differing

growth rates, ECM isolates were grown for nine weeks, *C. puteana* for five weeks and *S. lacrymans* for three weeks.

Biomass was harvested and washed twice in sterile double distilled water (ddH$_2$O) by adding 5 mL ddH$_2$O, briefly shaking

and centrifuging for 10 minutes at 1500 rpm (Beckman GS-6, California, USA). Following this, biomass of individual

replicates was freeze-dried, weighed and milled. To obtain enough biomass for elemental analysis replicates were then

pooled into 50 mL falcon tubes to give three samples per isolate per treatment. Ten 5 mm glass beads were added to each

tube, and they were loaded into a 50 mL tube adapter (Retsch MM 400, Haan, Germany) and milled for 15-30 minutes at 30

Hz until a fine powder was obtained.





### 2.2.3 Elemental analyses and statistics

Biomass samples were sent to the James Hutton Institute, Aberdeen, UK for elemental analyses. C and N analysis (Mettler

MT5 Microbalance, Mettler-Toledo GmbH, Laboratory & Weighing Technologies, Greifensee, Switzerland and Thermo

Finnigan Elemental Analyser FlashEA 1112 Series, Thermo Fisher Scientific, Massachusetts, USA), nitric acid microwave

digestion (PerkinElmer Titan MPS Microwave Sample Preparation System, PerkinElmer, Massachusetts, USA) and ICP-

EOS analysis (AVIO 500 ICP-OES, PerkinElmer, Massachusetts, USA) of Ca, Fe, K, Mg, Na and P was conducted on

fungal biomass, cellophane, gabbro and granite.


Individual biomass per plate and elemental concentration in biomass pools from each isolate and treatment were used for

statistical analysis, which was performed using the statistical software R v4.3.2 (R Core Team, 2023). The 'Tidyverse' suite

v2.0.0 was used for most data processing and visualisation (Wickham et al., 2019). Carbon and N measurements were used

to calculate C:N ratios. Linear models were fitted for each element to estimate the effects of isolate, treatment, and their

interactive effect on the individual response variables This was repeated for biomass and C:N ratio measurements, with

species in place of isolate. These effects were tested with the 'emmeans' v1.10.4 package (Lenth, 2024) and compact letter

displays were determined with the 'multcomp' v1.4-26 packages (Hothorn et al., 2008). This was followed by pairwise

comparisons of elemental concentration between treatments within each isolate, and biomass and C:N ratio between

treatments within each species and between species within each treatment, using Tukey's HSD. Weathering was defined as a

significantly higher element concentration after growth with gabbro and/or granite as compared to the limited treatment,

based on the linear models' output.

*Serpula lacrymans* and *C. puteana* were excluded from biomass analyses based on individual plates due to difficulties

harvesting complete biomass. The following isolates and treatments included fewer than three replicates for elemental

analyses due to a lack of biomass; *P. byssinum* and *S. lacrymans* all treatments (1 replicate), isolates Sl2 and Sl3 limited

analyses due to a lack of biomass; *P. byssinum* and *S. lacrymans* all treatments (1 replicate), isolates Sl2 and Sl3 limited

treatment (1 replicate), and Sl3 gabbro treatment (2 replicates).

To evaluate whether base cation uptake by *Suillus* growing with mineral additions was greater compared to other fungal

isolates linear models were fitted to estimate the differences in the ratio of individual element concentration in mycelia

growing in gabbro and granite treatments compared to the limited treatment, with isolate and treatment as explanatory

factors. This was tested with the 'emmeans' v1.10.4 package (Lenth, 2024) and results are reported as means with

confidence intervals.



Correlations between base cation transporter gene family copy numbers and means of mycelial elemental concentration for

all isolates in each mineral treatment and the control treatments were tested using Spearman's correlation. Base cation

transporter gene families were restricted to those which were reported as significantly expanding or contracting in the

CAFE5 analyses. The TCDB substrate search tool (https://tcdb.org/progs/?tool=substrate#/ ) was used to determine if any

base cation transporter gene families were specifically involved in the transport of Ca, Fe, K, Mg, Na and P. Isolates without

transporter family copy number data were excluded (*S. granulatus*, *S. grevillei*, *P.* aff. *fallax*, *Piloderma* sp. 1 and *Piloderma*

sp. 2).

## 3 Results

### 3.1 Phylogenetic Analyses

#### 3.1.1 Agaricomycotina phylogeny

The phylogenetic inference of 108 Agaricomycotina genomes was performed using 152 single copy orthologs that were

present in at least 50% of the taxa, which after concatenation resulted in a supermatrix with 54,895 amino acids. Most nodes

recovered a bootstrap value of 100 (Fig. S1).

#### 3.1.2 Transporter gene family copy numbers of 108 Agaricomycotina taxa

There are significantly ($p < 0.05$) more gene copy numbers of the major facilitator (MFS) gene superfamily (TCDB ID

2.A.1), the $Mg^{2+}$ transporter-E (MgtE) gene family (TCDB ID 1.A.26), the ATP-binding cassette (ABC) gene superfamily

(TCDB ID 3.A.1), the peroxisomal protein importer (PPI) gene family (TCDB ID 3.A.20) and the mitochondrial carrier

(MC) gene family (TCDB ID 2.A.29) compared to other transporter gene families (Fig. S2). Of these, the MSF gene

superfamily, the MgtE gene family, the ABC gene superfamily and the MC gene family are involved in base cation

transport. The MSF gene superfamily has notably high copy numbers in all 108 taxa, ranging from 61 to 294. The highest

copy number for a single species is in the MgtE gene family with 563 copies in *Tulasnella calospora*. Visualisation of

transporter gene family copy numbers by PCA showed a clear clustering of the families Suillaceae and Russulaceae which

are distinct from each other and all other taxa (Fig. 1a). Among the 30 most contributing transporter gene families, 18 are

base cation transporter gene families (Fig 1b).



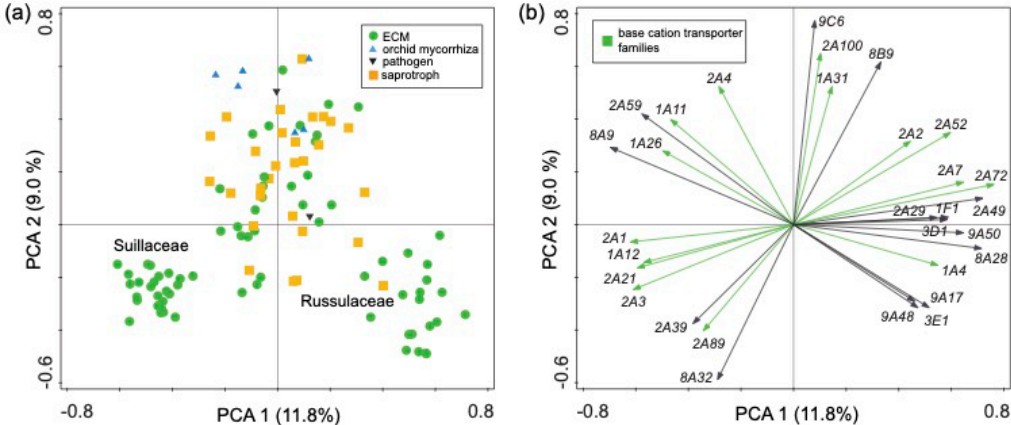


Figure 1 - Variation in transporter gene family composition in 108 Agaricomycotina taxa, based on Principle Component

Analysis of copy numbers of 173 transporter gene families and visualized by (a) sample and (b) species plots showing the 30

transporter gene families with the greatest contribution to the ordination of taxa. Lifestyle (a) is coded by shape and colour of

sample points, and in (b) base cation transporter families are indicated in green. The first two axes together explained 20.8%

of a total variation of 9586.4.

### 3.1.3 Transporter gene family evolution

Thirteen out of 114 analysed transporter gene families in the full dataset were found to have significant expansions and

contractions (Fig. S3-15), six of which are base cation transporter gene families and one of which is a base cation transport

accessory family (Tab. S2). In the partial family dataset, 22 of the 126 analysed transporter gene families had significant

expansions and contractions (Fig. S18-39), with nine being base cation transporter gene families and 2 being base cation

transport accessory families (Tab. S2). All significantly expanding or contracting transporter gene families in the full dataset

are in the genus *Suillus* (Fig. S3-15). Other clades with higher density of expansions and contractions are the families

Atheliaceae and Russulaceae (Fig. S16). In the partial family dataset, where 23 of the 28 taxa are *Suillus* species, there is a

loss of three and an addition of 12 transporter gene families which are expanding and/or contracting significantly in the

genus *Suillus*, when compared to the full dataset (Tab. S2). In the partial subfamily dataset, 15 of 51 analysed transporter

gene families were shown to be significantly expanding and contracting (Tab. S3). Of these, ten are involved in base cation

transport (Fig. S41-56, Tab. S3).



### 3.1.3.1 Base cation transporter gene family evolution

Within the genus *Suillus*, several branches show significant expansions and contractions in base cation transporter gene families in the full dataset (Fig S17), with the most frequent transporter gene families being the MgtE gene family (Fig. S3) and the MSF gene superfamily (Fig. S5). This pattern is reflected in the partial family dataset with 14 branches showing significant expansions and contractions in base cation transporter gene families (Fig. 2). The MgtE gene family (Fig. S19) is again one of the most frequent gene families, along with the β-amyloid cleaving enzyme 1gene family (TCDB ID 1.A.33) (Fig. S20). In the partial subfamily dataset, which included subdivisions of transporter gene superfamilies which were significantly expanding or contracting in the partial family dataset, there are significant expansions and contractions in branches leading to 14 internal nodes (Fig. S56), with the drug:$H^+$ antiporter-1 gene family (TCDB ID 2.A.1.2) (Fig. S42) and the pleiotropic drug resistance gene family (TCDB ID 3.A.1.205) (Fig. S50) as the most frequent gene families. There are significant expansions and contractions within the clade containing Suillus species in the full dataset (Fig S17), with the MSF gene superfamily (Fig. S5) and the ABC gene superfamily (TCDB ID 3.A.1) (Fig. S7) being the most frequent ones. In the partial family dataset, there are also many significant expansions and contractions in the clade containing *Suillus* species (Fig. 2). The most frequent gene families are again the MSF gene superfamily (Fig. S21) and the ABC gene superfamily (Fig. S26), with the addition of the MgtE gene family (Fig. S19) and the β-amyloid cleaving enzyme 1 gene family (Fig. S34). Within the genus the highest number of expansions and contractions in base cation transporter gene families in the full dataset occur in the branches leading to the ancestral node of *S. variegatus, S. tomentosus, S. hirtellus* and *S. fuscotomentosus*; and *S. cothurnatus, S. lakei,* and *S.* cf *subluteus* (Fig. S17). In the partial family dataset, *S. cothurnatus, S. lakei,* and *S.* cf *subluteus* again are inferred to have high numbers of expansions and contractions, as well as branches leading to the ancestral node of *S. placidus* and *S. weaverae/grevillei*; the ancestral node of *S. occidentalis, S. luteus* and *S. brevipes*; the ancestral node of *S.* cf *subluteus* and *S. subalutaceus*; and *S. decipiens, S. pictus/spraguei, S. plorans, S. subalutaceus, S. subaureus* and *S. variegatus* (Fig. 2).

Branches within the *Piloderma* clade also exhibit a number of expansions and contractions in both the full and the partial family datasets (Fig. S17 and Fig. 2, respectively) and, similar to *Suillus*, the most frequent base cation transporter gene family is the MgtE gene family (Fig. S3 and S19, respectively). *Piloderma byssinum* in particular has similarly high numbers of expansions in base cation transporters in both datasets (Fig. S17 and Fig. 2), and *P. olivaceum* shows high numbers of contractions in the partial family dataset (Fig. 2). Branches leading to the saprotrophic fungi *C. puteana* and *S. lacrymans* show low numbers of significant expansions and contractions in base cation transporter gene families in general, with *C. puteana* displaying one expansion in the partial family dataset (Fig. 2) and *S. lacrymans* displaying one contraction in the full dataset (Fig. S17) and two contractions and one expansion in the partial family dataset.



Other branches leading to nodes with high numbers of significant expansions and contractions in base cation transporter

gene families from the full dataset include those leading to the ancestral node of the genus *Lactarius*; the ancestral node for

*Russula vinacea*, *R. ochroleuca*, *R. rugulosa* and *R. emetica*; the ancestral node for *Paxillus involutus* and

*P. ammoniavirescens*; the ancestral node for *Fibulorhizoctonia psychrophila*, *Piloderma olivaceum*, *P. byssinum* and

*P. sphaerosporum*; and *Amanita rubescens*, *Gymnopus androsaceus, Fibulorhizoctonia psychrophila*, *Rhizopogon*

*vesiculosus*, *Pisolithus microcarpus*, *Melanogaster broomeianus*, *Paxillus involutus*, *Lactarius indigo*, *L. sanguifluus*,

*L. deliciosus*, *L. hengduanensis*, *L. pseudohatsudake*, *L. akahatsu* and *Ceratobasidium* sp. *anastomosis* (Fig. S17).



Figure 2 - Phylogenomic tree showing total of significant expansions and contractions of all base cation transporter gene families in the partial family dataset (28 taxa selected as they or their close relatives were present in the pure culture experiment). Branch-length not to scale.



### 3.1.3.2 Fungal mating-type pheromone receptor (MAT-PR) transporter gene family evolution

Expansions and contractions in transporter family genes related to sexual reproduction can have implications in the rate of
evolution and adaptation, which may lead to changes in mineral weathering capabilities of taxa. The fungal mating-type
pheromone receptor (MAT-PR) gene family (TCDB ID 9.B.45) was inferred to be significantly expanding in the branch
leading to the ancestral node of *S. occidentalis, S. luteus* and *S. brevipes*; and in *S. clintonianus*, *S. variegatus*, *S. tomentosus*
and *S.* cf. *subluteus* in the full dataset (Fig. S15). In the partial family dataset, significant expansions were found in the

branch leading to the ancestral node of *S. occidentalis, S. luteus* and *S. brevipes*; the branch leading to the ancestral node of
*S. plorans*, *S. variegatus*, *S. tomentosus*, *S. hirtellus* and *S. fuscotomentosus* and *S. discolor*; and the branch leading to the
ancestral node of *S. variegatus*, *S. tomentosus*, *S. hirtellus* and *S. fuscotomentosus*; in addition to *S. clintonianus*,
*S. variegatus*, *S. tomentosus*, *S. cothurnatus*, *S. pictus/spraguei*, *S. brevipes*, *S. occidentalis*, *S. placidus*, *S. ampliporus* and
*S.* cf. *subluteus* (Fig. S39). Significant contractions were inferred in the branch leading to *S. luteus* in the full dataset and in

those leading to *S. lakei*, *S. weaverae/grevillei*, *S. luteus*, *S. hirtellus*, *S. bovinus*, *S. decipiens* and *S. subaureus* in the partial
family dataset. Additionally, in the full dataset there is a large significant expansion in *R. Emetica*.

### 3.1.4 Mg$^{2+}$ transporter-E (MgtE) gene family evolution

A phylogenetic tree of protein sequences belonging to the MgtE gene family and present in the partial family dataset was

reconstructed in order to understand whether there has been an expansion of one homologous transporter gene in the genus
*Suillus*, or if there are several transporter genes of different origin. The phylogeny shows several clades of MgtE genes
present in each of the taxa (Fig. S57).

### 3.2 Base cation uptake by mycelia grown in pure culture

### 3.2.1 Correlation between base cation transporter gene family copy number and mycelial base cation concentration

Of the 25 base cation transporter gene families found to be significantly expanding and/or contracting, a total of six base
cation transporter gene families that are involved specifically in the transport of Ca, Fe, K, Mg, Na and P, and five other base
cation transporter gene families which are more generally reported as base cation transporting without being specifically

linked to transport of a particular base cation, showed significant correlations between gene family copy number and base
cation uptake. There was a total of 28 significant correlations between mycelial concentration of base cations and copy
numbers of base cation transporter gene families across species within the different treatments. Most of the significant
correlations were positive with increasing copy numbers resulting in larger base cation uptake, but some were also negative
(Fig. 3 and S58-68).






Mycelial Mg concentration was significantly positively correlated with genome copy numbers of the MgtE gene family in the gabbro, granite and limited treatments, and there was a positive correlation in the rich treatment (Fig. 3a-d, respectively). There was a significant positive correlation in the limited treatment for Ca concentration (Fig. 3g), no correlations in the gabbro and the granite treatments and a negative correlation in the rich treatment (Fig. 3h). There were also negative

correlations in all treatments for Fe concentration (Fig. 3i-l), though not significant.

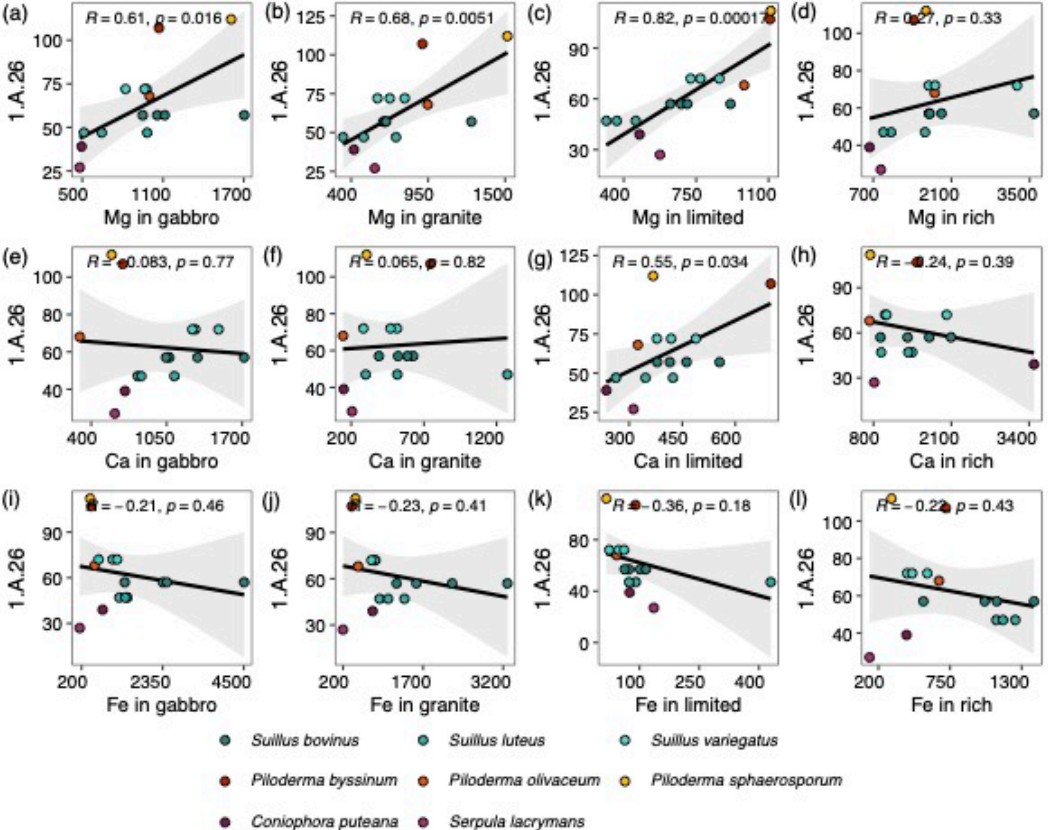

Figure 3 - Correlations between mycelial magnesium (Mg), calcium (Ca) and iron (Fe) concentrations (mg/kg) and transporter gene copy numbers. Element concentration is given as the estimated mean values in eight different fungal species grown in pure culture in gabbro, granite, limited and rich treatments, and the gene copy numbers of the Mg2+ transporter-E

family (TCDB ID: 1.A.26) in the corresponding fungal genomes.



Magnesium mycelial concentration is also significantly positively correlated with copy numbers of the transient receptor

potential $Ca^{2+}$/cation channel (TRP-CC) gene family (TCDB ID 1.A.4) (Fig. S58k), the ABC gene superfamily (Fig. S59g)

and the ankyrin gene family (TCDB ID 8.A.28) (Fig. S60o) in the limited treatment, and of the P-type ATPase (P-ATPase)

gene superfamily (TCBD ID 3.A.3) in the granite treatment (Fig. S61n); however there is a significant negative correlation

with copy numbers of the $Ca^{2+}$, $Mn^{2+}$- ATPase gene family (TCDB ID 3.A.3.2) in the limited treatment (Fig. S62k). In the

rich treatment, mycelial Na concentration is significantly positively correlated with copy numbers of the anion:cation

symporter (ACS) gene family (TCDB ID 2.A.1.14) (Fig. S63h), the $H^+$- or $Na^+$-translocating F-type, V-type and A-type

ATPase (F-ATPase) gene superfamily (TCDB ID 3.A.2) (Fig. S64d), the $Ca^{2+}$, $Mn^{2+}$- ATPase gene family (Fig. S62p), the

nucleobase:cation symporter-1 gene family (TCDB ID 2.A.39) (Fig. S65h), and the G-protein-coupled receptor (GPCR)

gene family (TCDB ID 9.A.14) (Fig. S66d); and is significantly negatively correlated with copy numbers of the TRP-CC

gene family (Fig. S58p). In the gabbro treatment, mycelial Na concentration is also significantly positively correlated with

copy numbers of the GPCR gene family (Fig. S66c), and in the gabbro and the limited treatments with the β-amyloid

cleaving enzyme 1 (BACE1) gene family (TCDB ID 8.A.32) (Fig. S67q and S67t, respectively). Potassium mycelial

concentration is significantly positively correlated with copy numbers of TRP-CC gene family (Fig. S58g) and the P-ATPase

gene superfamily (Fig. S61k), and negatively correlated with copy numbers of the ACS gene family (Fig. S63c) in the

limited treatment. Calcium mycelial concentration is significantly positively correlated with copy numbers of the amino

acid-polyamine-organocation (APC) gene superfamily (TCDB ID 2.A.3) in the gabbro (Fig. S68a) and the limited (Fig.

S68c) treatment, the nucleobase:cation symporter-1 gene family in the gabbro treatment (Fig. S65a), the ankyrin gene family

in the limited treatment (Fig. S60c), and the BACE1 gene family in the gabbro treatment (Fig. S67a). Mycelial Ca

concentration is significantly negatively correlated with copy numbers of the TRP-CC gene family (Fig. S58a) and the

ankyrin gene family (Fig. S60a) in the gabbro treatment. There are significant positive correlations between Fe mycelial

concentration and copy numbers of the BACE1 gene family in the gabbro (Fig. S67e) and granite (Fig. S67f) treatments.

There are no significant correlations between P concentration and copy numbers of P transporting transporter gene families,

however, correlations are generally positive regardless of treatment (Fig. S59 and S63).

### 3.2.2 Mycelial base cation concentration

For mycelial concentration of each base cation measured there were significant ($p < 0.001$) overall effects of isolate,

treatment (gabbro, granite, limited or rich) and the interaction between isolate and treatment. For all elements there were

examples of higher element concentrations after growth with gabbro and/or granite as compared to the limited treatment.

For example, all *S. bovinus* isolates showed significantly greater Mg mycelial concentration in the gabbro treatment

compared to the granite and limited treatment (Fig. S69d). Additionally, one *S. luteus* and one *S. variegatus* isolate, and the



*Piloderma* isolates Pf and Ps showed significantly greater Mg mycelial concentrations in the gabbro treatment compared to the limited (Fig. S69d). For Fe, all isolates excluding *P. byssinum* and *S. lacrymans* showed significantly greater Fe mycelial

concentrations in the gabbro and the granite treatment compared to the limited (Fig. S69b). For Ca, all *S. bovinus*, *S. variegatus*, two *S. luteus*, one *S. granulatus* and the saprotroph *C. puteana* showed significantly greater mycelial concentrations in the gabbro treatment compared to the limited (Fig. S69a). Two *S. bovinus* isolates and two *Piloderma* isolates showed significantly greater K mycelial concentrations in the gabbro treatment compared to the limited (Fig. S69c), and one *S. bovinus* and one *S. luteus* isolate showed significantly greater P mycelial concentrations in the gabbro compared

to the limited treatment (Fig. S69f). All *Suillus bovinus* isolates had significantly greater Na mycelial concentrations in the gabbro and limited treatments compared to the rich treatment, and one *Piloderma* isolate showed a significantly greater Na mycelial concentration in the gabbro treatment compared to the limited treatment (Fig. S69e).

There were also examples of higher element concentration in the rich treatment as compared to the mineral treatments.

Nearly all isolates showed significantly greater Ca, Fe and Mg mycelial concentrations in the rich treatment compared to the limited (Fig. S69d). Six *Suillus* isolates and four *Piloderma* isolates showed significantly greater K mycelial concentrations in the rich treatment compared to the limited (Fig. S69c). For Na, four *Suillus* isolates showed significantly lower concentrations and one *S. luteus* isolate showed a significantly higher concentration in the rich compared to the limited (Fig. S69e) For P concentrations, eight *Suillus* isolates and four *Piloderma* isolates showed significantly higher concentrations in

the rich compared to the limited treatment (Fig. S69f).

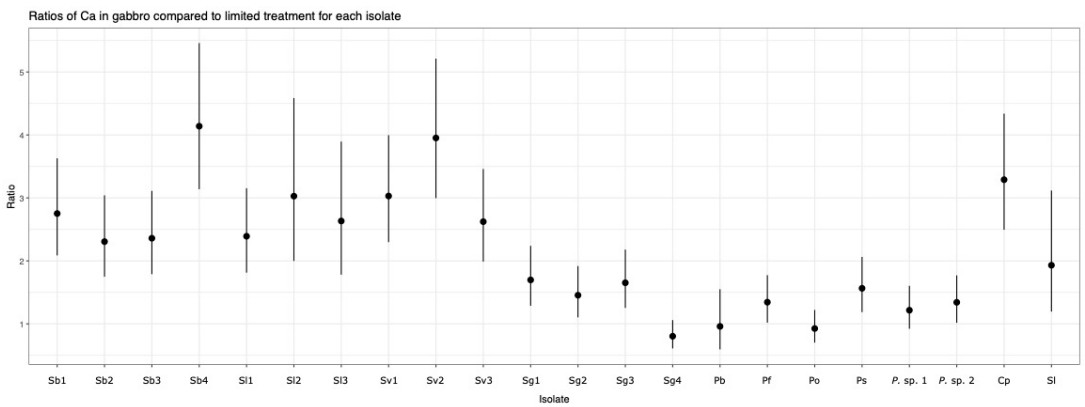

Figure 4 - Ratios of calcium (Ca) concentration (mg/kg) between gabbro and limited treatments for each fungal isolate.

Isolates include *Suillus bovinus* (Sb1-4), *S. luteus* (Sl1-3), *S. variegatus* (Sv1-3), *S. granulatus* (Sg1-3), *S. grevillei* (Sg4),

*Piloderma byssinum* (Pb), *P.* aff. *fallax* (Pf), *P. olivaceum* (Po), *P. sphaerosporum* (Ps), *Piloderma* sp. 1, *Piloderma* sp. 2,



*Coniophora puteana* (Cp), and *Serpula lacrymans* (Sl). Data points represent the mean ratio between Ca concentration in gabbro and in limited for each isolate, with error bars showing the confidence interval.

Comparisons between *Suillus, Piloderma*, and saprotroph isolates of the ratio of individual element concentrations in mycelia grown in mineral treatments compared to the limited treatment showed different patterns (Fig S70, S71). The linear

models showed many instances of a significant overall effect ($p < 0.05$) of isolate and treatment on the ratio between concentrations of each element in the mineral and limited treatments. The strongest difference between groups was seen for Ca in the gabbro treatment where linear models showed the ratio between gabbro and limited were significant for all isolates except for one *Suillus* and four *Piloderma* isolates; and the ratios for *Suillus* isolates in general were higher compared to *Piloderma* isolates, and in many cases significantly higher (Fig 4). In the granite treatment the pattern for Ca ratios was

similar but weaker, where ratios for only four *Suillus* isolates and one *Piloderma* isolate were significant from the linear models (Fig. S71a). For Fe the isolates showed a similar pattern in ratios in both gabbro and granite, including higher ratios for all *S. bovinus* isolates, two *S. variagatus* isolates, and *Piloderma sphaerosporum* (Fig. S70a and S71b, respectively). This was reflected in the linear models, where only the ratio of the saprotroph *S. lacrymans* was not significant in either the gabbro or the granite treatments. For Mg in the gabbro treatment some *Suillus* isolates had significantly higher ratios

compared to some of the *Piloderma* isolates, but *Piloderma sphaerosporum* also had a high ratio (Fig. S70c). In granite the Mg ratios showed less pronounced differences between isolates (Fig. S71d). In the gabbro, the linear models showed that Mg ratios in 11 *Suillus* and two *Piloderma* isolates were significant, and in the granite Mg ratios of only four *Suillus* isolates and one *Piloderma* were significant. The ratios for K, Na and P mostly did not differ between isolates (Fig. S70b and S71c, S70d and S71e, and S70e and S71f, respectively) and linear models showed very few instances of significant ratios in the gabbro

and the granite treatments.

### 3.2.3 Mycelial biomass and C:N ratio

There was a significant overall effect ($p < 0.001$) of species, treatment (gabbro, granite, limited or rich), and their interaction for fungal biomass and C:N ratio. Overall, *Suillus* and *Piloderma* species produced similar amounts of biomass across all

treatments (Fig. S72). *Piloderma olivaceum* had significantly higher biomass than all other species in the gabbro treatment, followed by *S. variegatus* which had significantly higher biomass compared to the remaining species. In the granite and rich treatments *P. olivaceum* and *S. variegatus* both had significantly greater biomass than other species, whilst *S. variegatus*



alone had significantly greater biomass compared to other species in the limited treatment. *Piloderma byssinum* produced significantly less biomass than all other species in all treatments.


For comparisons between treatments within each species *Suillus bovinus* showed a significant increase in biomass in the rich treatment compared to all other treatments (Fig. S73). *Suillus luteus*, *P. olivaceum* and isolates *Piloderma* sp. 1 and sp. 2 produced significantly more biomass in the gabbro, granite and rich treatments compared to the limited treatment, with *Piloderma* sp. 2 producing significantly more biomass in the rich treatment compared to the granite treatment. *Suillus*

*grevillei* produced significantly greater biomass in the granite treatment compared to the limited and *P. byssinum* had significantly higher biomass in the rich treatment compared to the limited.

When comparing C:N ratios between treatments within species *S. luteus* is the only species which displayed a significant difference between treatments, with higher C:N ratio in the rich treatment compared to the limited (Tab. S5a), while all other

species were similar. When comparing between species within treatments, *Suillus* species generally had a higher C:N ratio than *Piloderma* species, and saprotrophic species generally had low C:N ratios (Tab. S5b). In particular, *S. variegatus* had a significantly higher C:N ratio than all other species excluding *S. luteus* and *S. granulatus* in the gabbro treatment. This pattern was repeated in the other treatments, excluding *S. granulatus* in the granite and limited treatments, and including *P. olivaceum* in the rich treatment.

**4 Discussion**

**Rapid base cation transporter gene family evolution in *Suillus* species**

The requirement for base cations by plants and fungi is a key driver of biological mineral weathering and a number of ECM fungal species have been shown to take part in this process by colonising mineral surfaces, producing LMWOAs and altering soil chemistry (Calvaruso et al., 2013; Gazze et al., 2012; Griffiths et al., 1994; Schmalenberger et al., 2015; van Scholl et

al., 2006). The genus *Suillus* is a group of ECM fungi with high host specificity (Lofgren et al., 2021) and the capability to take up base cations from mineral sources (Balogh-Brunstad et al., 2008a; Wallander, 2000). In the ordination based on copy numbers of all transporter gene families, members of the families Suillaceae and Russulaceae clustered separately from each other and other Agaricomycotina species (Fig. 1), and among the most contributing transporter gene families, base cation transporter gene families were highly abundant indicating that this group of transporters may be important in the lifestyle

and environmental adaptations of these two groups.



As a general rule, copy numbers of genes related to cell and organism maintenance and growth remain relatively constant as mutations in these genes would likely reduce fitness, however, copy numbers of genes related to ecological adaptations, including stress, have greater copy number variation as changes in these genes are less likely to reduce fitness and may give

rise to advantageous phenotypes (Wapinski et al., 2007). It is therefore expected that transporter genes, which frequently have roles in abiotic and biotic environmental interactions, will display greater variation in copy numbers over time, as a result of expansions and contractions, and that they have important roles in the rapid adaptation to environmental changes and stress. Our analyses of 108 Agaricomycotina genomes confirm significant expansions and contractions in base cation transporter gene families and display a rapid evolution of these transporter gene families in the genus *Suillus*, thereby

supporting our first hypothesis that greater base cation uptake is dependent on evolutionary expansions in copy numbers of base cation transporter gene families.

Thirteen transporter gene families in the full dataset, 22 in the partial family dataset and 15 in the partial subfamily dataset were found to have significant expansion and contractions, all of which have significant expansions and contractions in the

*Suillus* genus. Differences in the number of transporter families with significant expansions and contractions between datasets are likely due to the scaling of the full dataset and the difference in the number of taxa included. These significant expansions and contractions, suggest that these gene families have rapidly evolved in the genus *Suillus* enabling species within this group to adapt to their environment.

There are also significant expansions in the fungal mating-type pheromone receptor (MAT-PR) family in a number of *Suillus* species, in both the full and partial family datasets, and a large expansion in *Russula emetica* in the full dataset. This transporter gene family is reported to play a role in chemotrophic sensing of host plants (Turrà et al., 2015), and in female sexual development in *Neurospora crassa* (Krystofova and Borkovich, 2006). Sexual reproduction is often favoured in unstable environments or in situations where sources of nutrients may be inconsistent (Nieuwenhuis and James, 2016), and

adaptations in sexual populations can occur faster than in asexual populations (Becks and Agrawal, 2012; McDonald et al., 2016; Nieuwenhuis and James, 2016)  Therefore, expansions in this transporter family may indicate an increase in the proportion of sexual reproduction performed relative to asexual reproduction, which likely results in populations more rapidly adapting to environmental changes.



**Significantly evolving base cation transporters and their functions**

Approximately half of the significantly expanding and contracting transporter gene families found in each dataset are involved in base cation transport, indicating that transport of base cations may be important in adjusting to environmental fluctuations and stress, as genes which vary in copy number are more likely to have roles in adaptation (Wapinski et al., 2007). In addition, expansions and contractions in these transporter families are likely to be related to the requirement of base cation uptake, and their occurrence within taxa, such as the genus *Suillus*, may be an indication of mineral weathering capabilities. Branches leading to *S. lakei, S. variegatus, S. cothurnatus* and *S.* cf *subluteus,* as well as two internal branches within the genus *Suillus* possess significant expansions in the MgtE gene family (Fig. S19). Though primarily studied in bacteria and mammals, members of the MgtE gene family are identified as facilitators of $Mg^{2+}$, $Zn^{2+}$, $Co^{2+}$, $Ni^{2+}$, Fe and Cu influx (Goytain and Quamme, 2005; Smith et al., 1995), and play roles in $Mg^{2+}$ homeostasis (Conn et al., 2011; Franken et al., 2022; Hattori et al., 2009; Hermans et al., 2013). Additionally, it has been found that many organisms have multiple copies of one type of $Mg^{2+}$ transporter gene family, which may explain the relatively low abundance of other $Mg^{2+}$ transporter gene families in our datasets (Franken et al., 2022). In plants, it was reported that there are many gene copies of a $Mg^{2+}$ transporter gene family, the CorA/Mrs2 metal ion transporter (MIT) gene family, and that they are not functionally redundant (Gebert et al., 2009; Schmitz et al., 2013). Our analysis suggests multiple different homologous MgtE genes have undergone expansions, as observed in the phylogenetic gene tree of MgtE gene family members present in the partial dataset (Fig. S57), so rather than having a not functionally redundant transporter, there might be multiple homologues performing the same function or each with with a different function. In the genus *Suillus*, the major facilitator gene superfamily has six significant expansions in both the full and the partial family datasets, three and nine significant contractions in the full and partial family datasets, respectively, and one significant contraction in *P. sphaerosporum* in both datasets (Fig. S5, Fig. S21). This gene superfamily is large and diverse and reported to transport solutes such as sugars and cations (Law et al., 2008).

Although *S. bovinus* possesses two significant contractions in the full dataset, three in the partial family dataset and no expansions in base cation transporter families, it had significantly higher mycelial concentrations of Mg, Ca and Fe in the gabbro treatment compared to the limited treatment. One isolate in particular (Sb1) had significantly higher mycelial concentration of all base cations in the gabbro treatment compared to the limited. This corroborates previous findings, of $Mg^{2+}$ uptake from B horizon soil solution, where *S. bovinus* was the most abundant species (Mahmood *et al.*, 2024). Additionally, *S. bovinus* showed a clear increase in abundance from the O to E to B horizon, a pattern that was repeated in the in-growth mesh bags which were placed in the horizons, indicating the preference of *S. bovinus* toward mineral soils (Mahmood et al., 2024).


*Suillus luteus* has no significant expansions or contractions in the full dataset and one contraction in the full and partial family datasets, but had significantly higher mycelial concentrations of Ca and Fe in the gabbro treatment compared to the limited treatment. These findings confirm our hypothesis about mineral weathering resulting in base cation uptake when *S, bovinus* and *S. luteus* grow with gabbro additions. Notably, isolate Sl2 had significantly higher mycelial concentrations of

Mg, Ca, Fe and P in the gabbro treatment compared to the limited treatment. *Suillus luteus* has previously been reported to be tolerant to heavy metal exposure, specifically zinc and cadmium which are both cations (Bazzicalupo et al., 2020; Colpaert et al., 2011; Ruytinx et al., 2011, 2017, 2019). These adaptations were concluded to be due to single nucleotide polymorphisms and gene copy number variation in genes involved in heavy metal homeostasis, which includes metal ion transporter genes, in tolerant populations (Bazzicalupo et al., 2020). These findings align with our first hypothesis that

adaptation to the environment, such as greater uptake of base cations, can occur through expansions in gene family copy number (Bazzicalupo *et al.*, 2020).

In addition to significant expansions in the MgtE gene family, *S. variegatus* also possesses significant expansions in the nucleobase:cation symporter-1 gene family (TCDB ID 2.A.39) which is associated with Na uptake. In pure culture, *S.*

*variegatus* had significantly higher mycelial Ca and Fe concentrations in the gabbro compared to the limited treatment, further confirming the hypothesis about mineral weathering resulting in base cation uptake when fungi are growing with mineral additions. Previously, *S. variegatus* has been shown to take up $K^+$ from granite particles (Fahad et al., 2016), to produce high levels of organic acids in the presence of minerals (Olsson and Wallander, 1998; Wallander and Wickman, 1999) and to demonstrate raised levels of weathering when in symbiosis with *Pinus sylvestris* compared to non-mycorrhizal

plants (Wallander and Wickman, 1999), ratifying our second hypothesis. In addition to significant expansions in the MgtE gene family, *S.lakei*, *S. cothurnatus* and *S.* cf *subluteus* have particularly high numbers of other base cation transporter gene families with significant expansions in the full and the partial family datasets, which may also indicate a heightened ability to take up base cations and potentially weather minerals, however this needs experimental confirmation.

**Proportionally higher base cation uptake of Ca in *Suillus* compared to *Piloderma* indicating greater weathering capabilities**

Whether base cation uptake by *Suillus* species growing with mineral additions was greater compared to *Piloderma* species was partly confirmed by comparing the differences in the ratio of element concentrations in the mineral treatments compared



to the nutrient limited media. Only Ca showed a clear pattern for *Suillus* as a group, but other base cations either showed no

pattern or mixed patterns. Taken together this may indicate a greater weathering capability for *Suillus*, especially in the

gabbro, which has a higher proportion of weatherable minerals compared to the granite. Though it was expected that *Suillus*

isolates would take up more base cations in pure culture than all other isolates, a number of *Piloderma* isolates performed

equally well (Fig. S69). Similarly to *Suillus* species, *P. byssinum* and *P. sphaerosporum* had significant expansions in the

MgtE gene family (Fig. S3 and S19), indicating that these isolates also are capable of not only taking up base cations, but

potentially also of weathering minerals.

**Mycelial base cation uptake correlates with base cation transporter gene family copy numbers**

We found 28 significant correlations between different elements and copy numbers of base cation transporter gene families

(which were significantly expanding or contracting) associated with their uptake, confirming the hypothesis that base cation

uptake correlate with base cation transporter gene family copy numbers. For example, copy numbers of the MgtE gene

family were shown to have significant positive correlations with Mg uptake in the gabbro, granite and limited treatments and

a weak positive correlation in the rich treatment, which may indicate that these genes are only transcribed in conditions

where nutrients are limited and mineral weathering is required to access nutrients. The major facilitator gene superfamily

showed no significant correlation between gene copy number and base cation uptake, however when further separated into

transporter gene families, copy numbers of the anion:cation symporter (ACS) gene family, associated with K, Na and P

transport (Farsi et al., 2016; Pavón et al., 2008; Prestin et al., 2014), had a significant positive correlation with Na uptake in

the rich treatment. Not all transporter gene families associated with transport of a particular base cation showed a significant

positive correlation between gene copy number and uptake of that base cation, for example the TRP-CC gene family and the

ACS gene family showed  significant negative correlations with Ca uptake in the gabbro treatment and t K uptake in the

limited treatment, respectively, which may indicate that another mechanism of regulation, such as increasing rates of

transcription of a smaller number of genes, may be favoured. Additionally, there were significant correlations between

uptake of specific base cations and gene copy numbers of transporter families which are not associated with the transport of

those base cations. For example, the oligopeptide transporter (OPT) family, which is associated with Fe transport (Kurt,

2021; Mousavi et al., 2020; Wintz et al., 2003), was significantly negatively correlated with Mg uptake in the gabbro, granite

and limited treatments and may indicate an inhibitory effect of proteins of this family on Mg uptake (data not shown).

**Conclusions**

*Suillus* species which often grow in the B horizon in boreal forests have the capability to take up base cations from mineral sources, and here we tested whether greater base cation uptake is dependent on evolutionary expansions in copy-numbers of

base cation transporter gene families. Base cation transporter gene families were among the most abundant type of transporter gene families, with a rapid evolution in the genus *Suillus* in particular. This suggests that the expansions and/or contractions in these transporter gene families are likely related to the requirement of base cation uptake, underpinned by the finding that mycelial uptake of elements in the presence of gabbro and granite, two commonly occurring mineral-rich rock types, often increased with base cation transporter gene family copy numbers. Since the result was not confined to the genus

*Suillus* alone, but also detected for the genus *Piloderma*, some species of which are also found growing in mineral soil layers, this type of study using controlled experiments with ECM fungi growing in pure culture, or in symbiosis with a seedling, to confirm the weathering process should be expanded to include more species. Further work into other levels of regulation, e.g. through transcriptional regulation of transporter proteins, would also be useful to gain a deeper understanding of base cation uptake regulation and its role in mineral weathering by *Suillus* species, and ECM fungi in general.


**Supplementary Information**

Supplementary information is available on FigShare doi: 10.6084/m9.figshare.28016633.

**Author Contribution**

All authors contributed to conceptualisation of the project. KK and MSG performed bioinformatic analyses. KK and PF carried out experimental work and elemental analyses. KK wrote the manuscript with contribution from all co-authors.

**Competing interests**

The authors declare that they have no conflicts of interest


**Acknowledgements**

We thank Erica Packard and Carles Castaño Soler for assistance with collecting *Suillus* fruitbodies, Geoffry Daniels for use of fungal cultures, and Yasaman Najafi, Katarina Ihrmark, and Rena Gadjieva for help in the laboratory with the pure culture experiment.


**Financial support**

Swedish Research Council FORMAS (2020-01100).



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
