# Peer review of "Transporter gene family evolution in ectomycorrhizal fungi in relation to mineral weathering capabilities"

_EGUsphere, 2024_

## Author Comment (AC1)

Reviewer 1:

**General comments**

1. How are hypotheses 1 and 4 different? It makes sense to combine them: "Greater base cation uptake is dependent on base cation transporter gene family copy numbers that result from evolutionary expansions". This would require some reorganization of the results and discussion, but would them clearer and more concise.

We have combined hypothesis 1 and 4 as suggested (see abstract and introduction, lines 20-24 and 118-123) and altered throughout the manuscript accordingly.

*"We hypothesised that 1) greater base cation uptake is dependent on base cation transporter gene family copy numbers that result from evolutionary expansions, 2) mineral weathering results in base cation uptake by* Suillus *growing in pure culture with mineral additions, and 3) base cation uptake by members of the genus* Suillus, *frequently reported to perform mineral weathering, will be greater when growing in pure culture with mineral additions compared to other fungal species. A correlation between base cation uptake and base cation transporter gene family copy numbers would provide a mechanistic explanation for mineral weathering."*

2. Seventy-three supplemental figures is far too many. The authors should condense the expansion/contraction information from supplemental tree figures S3-S17 (which all appear to use the same phylogenomic tree) into one figure with a heatmap next to the tree, since they all represent the same idea. Likewise with figures S18-S56 for the partial dataset. The total number of gene family expansions/contractions shown in Fig 2 could also be included in this figure. This can be done in anvio by running anvi-import-misc-data for each tip of the tree.

Thank you for the good suggestion. We have condensed the supplementary figures S3-S17 (now S3), S18-S40 (now figure 3), and S41-S56 (now S4) into three figures, each with one summary tree and heatmaps showing the copy number change for each transporter gene family.

3. The gene family evolution results are basically presented as a list, and could be made much more concise. I suspect that condensing and synthesizing gene expansion/deletion data from supplemental figures would lead to a broader-scale presentation of results that reads more naturally. Same idea for the mycelial cation correlation results.

Overall, the manuscript is very long. Significant sections require editing or removal to be made more concise or clear—more is not always better. I've given some examples below.

We have made significant changes to reduce the length of the manuscript, the specific changes suggested by the reviewer were addressed and indicated below.

4. **Introduction**

a. Rewrite run-on or unclear sentences in lines 52-55, 57-60, and condense section from lines 80-91.

We have altered the text to make it clearer (see lines 56-59 – previously 52-55):

*"Base cation mobilisation through mineral weathering by ECM fungi primarily takes place in the E and B horizons (van Breemen et al., 2000; Jongmans et al., 1997; Mahmood et al., 2024). While fine root density in these horizons is lower than in the O horizon, the extensive mineral soil volume hosts up to two-thirds of ECM root tips, nearly half of all ECM taxa found across a boreal forest profile, and numerous ECM taxa that do not occur in other soil layers (Rosling et al., 2003)."*

(see lines 61-64 – previously 57-60):

*"Mahmood et al. (2024) demonstrated that weathering in the mineral soil is driven by C sinks created through ECM-mediated N mobilisation from the O horizon. Additionally, variations in organic matter availability in the O horizon were found to influence fungal C allocation to the E and B horizons, thereby altering the abundance of different functional fungal guilds."*

(see lines 89-97 – previously 80-91):

*"In the presence of minerals, several* Suillus *species have been observed to increase production and exudation of LMWOAs (Adeleke et al., 2012; Olsson and Wallander, 1998; Wallander and Wickman, 1999), and take up base cations such as potassium (K), Mg and iron (Fe) (Balogh-Brunstad et al., 2008a); Fahad et al., 2016). Mahmood et al. (2024) found that Mg2+ was taken up from B horizon soil solution in reconstructed podzol microcosms containing Pinus sylvestris seedlings growing in natural forest soils, where S. bovinus was the most abundant taxon. Other ECM fungi involved in mineral weathering include* Paxillus involutus *and* Pisolithus tinctorius *(Bonneville et al., 2009); Gazze et al., 2012; Van Scholl et al., 2006; Paris et al., 1995, 1996; Balogh-Brunstad et al., 2008b). However, previous studies primarily relied on abundance data and chemical analyses of ECM fungal biomass, plant biomass, or growth medium of experimental systems, without specifically investigating the mechanisms or regulation of base cation uptake."*

5. **Methods**

a. Line 220: I assume you meant micrometers instead of mm.

Yes, thank you for pointing that out, it has now been corrected (see line 242)

b. Line 246: How was biomass harvested? A brief description or citation would be sufficient.

We have added a brief description (see lines 250-251):

*"A sterile cellophane sheet, rinsed in deionised water and autoclaved, was placed on each*

*plate to prevent mycelial growth into the agar and to aid harvesting, whilst allowing the transfer of soluble compounds."*

*(see lines 258.259):*

*"Both control treatments were adjusted to pH 5.5 and covered with a sterile cellophane sheet."*

*(see line 266-267):*

*"Biomass was harvested by gently removing mycelia from the cellophane and the agar plug..."*

     c. Commas are needed to provide clarity on lines 160, 192, 279, and elsewhere in the manuscript. A period is missing in line 265.

Thank you for pointing this out. Commas have now been added to improve readability and the full stop has been added.

     d. Table 1: Why not simply specify the absolute concentrations of each mineral in each medium, rather than as a relative increase from the minimum medium?

The concentration of these elements is zero in the limited treatment, and we now added the limited treatment to the table to make this clear.

**Results/Figures/Supplementals**

     a. Line 300: Replace "There are" with "In the clade *Agaricomycotina*, we identified…"

We have changed "There are" to "In the clade Agaricomycotina, we identified". See line 320

     b. Line 305: Assume you meant MFS instead

Thank you for pointing this out, we have now corrected "MSF" to "MFS".

     c. Figure 1a: Curious to know if you observe a separation between the non-Suillaceae, non-Russulaceae ECM and the non-ECM if you plotted on one of the other primary axes. Do *all* ECM fungi have different transporter gene composition from non-ECM fungi?

No, they do not have completely different transporter gene compositions from non-ECM fungi, but good question!

     Similar comment for the correlations (Figures S58-S68), but do something like plot the R2 values to condense these into a single figure.

We tried condensing the plots by having one plot per transporter gene family per element with all four treatments in one, but found it to be visually too noisy. We instead opted to condense the supplementary figures into a table with only the R values as suggested and an asterisk indicating a significance of p ≤ 0.05, please see Table 2 on line 440

*"Table 2 – Summary of R values for each of the correlations between estimated mean mycelial elemental concentration and base cation transporter gene family copy numbers of eight different fungal species grown in pure culture in gabbro, granite, limited and rich treatments. Element, Treatment and TCDB IDs of base cation transporter gene families are indicated. R values with significant p values (p ≤ 0.05) are indicated in bold and with an asterisk. Correlations were not performed for cases where a base cation transporter gene family is not associated with the transport of the respective element, as indicated by empty cells."*

      c.   Fig. 2: Include in the legend that + refers to an expansion and – a contraction (I assume)

We have modified the figure and figure legend (now figure 3, and now with the heatmaps as well) (see lines 380-390):

*"Figure 3 - Summary of copy number changes in significantly expanding and contracting transporter families (TFs) across the partial dataset. The number of TFs expanding and contracting at each node of the phylogenomic tree (not to scale) are indicated with a positive (+) number for expanding families or a negative (-) number for contracting families. The heatmaps show changes in gene copy number of significantly expanding and contracting TFs, with contractions in blue and expansions in red at each node (left heatmap) and tip (right heatmap). Significant expansions and contractions are indicated with an asterisk (\*). The heatmap for internal nodes (left) has gene copy number changes ranging from -22 to +49, and dashed red lines indicating which node corresponds to with row of the heatmap. The heatmap for tips (right) has gene copy number changes ranging from -48 to +101. For both heatmaps, base cation TFs are positioned to the left and separated from other TFs which are positioned to the right."*

      d.   Fig. 3: Why show (or even mention, line 408) all the non-significant correlations? It distracts from the correlations you did find.

We have now removed the mention of the non-significant correlations in the gabbro and granite treatments, referred to the above-mentioned table of R values and highlighted Mg uptake and the MgtE gene family (see lines 428-429):

*"Significant correlations are indicated in Table 2. In particular, mycelial Mg concentration was significantly positively correlated with genome copy numbers of the MgtE gene family in the gabbro, granite and limited treatments, and there was a positive correlation in the rich treatment (Fig. 4a-d, respectively)."*

      e.   Line 444: Specify statistical test and results. Was this an ANOVA?

We have rephrased the text in the materials and methods, and in the results for clarity (see lines 284.286 – previously 264-265):

*"Linear models were used to perform ANOVA for each element, biomass, and C:N ratio to assess the effects of isolate (or species in the case of C:N ratio), treatment, and their interactive effect on individual response variables."*

*(see lines 447-448 – previously 443-444):*

*"For mycelial concentration of each base cation measured, ANOVA showed that there were significant (p < 0.001) overall effects of isolate, treatment (gabbro, granite, limited or rich) and the interaction between isolate and treatment."*

6. **Discussion**

   a. Line 535: This point would be made more strongly if presented after synthesizing the base cation uptake results.

We have now added the point of base cation uptake levels in the beginning of the discussion, to support hypothesis 1. In addition, we moved up the section about discussing correlation directly after "Significantly evolving base cation transporters and their functions". (see lines 516-518 and 586-603)

*"…Additionally, we identified 28 significant correlations between different elements and the copy numbers of expanding or contracting base cation transporter gene families associated with their uptake. **These findings support our first hypothesis that greater base cation uptake is driven by evolutionary expansions of base cation transporter gene families."***

   b. Line 540: How exactly might the choice or number of taxa influence the numbers of gene families found to have significant expansions/contractions, and what consequence might this have for interpreting these results? Please cite an example if possible.

In a larger dataset many more families might be included since they can be present at the root but not in a more exclusive dataset, for example family IA6 and 3D3 are present in the Agaricomycotina dataset, but not in the 28-taxa dataset. CAFÉ5 uses a birth-death process to model evolutionary expansions and contractions, when a family is present in some of the taxa but inferred to be absent from the root then it is excluded from the analyses as a family cannot "die" if it has not been "born", therefore when having a more inclusive dataset more families can be included. With this reasoning in mind, we decided to analyse the different datasets to identify some contractions/expansions that might be missing due to the size and membership of the different datasets.

   Line 580: Add a reference to a figure here.

Figure S6 has now been referenced here.

   c. Awkward/unclear phrasing: lines 582-584

We have altered the text to make it clearer (see lines 560-561 – previously 580-584):

*"This supports previous findings of Mg2+ uptake from B horizon soil solution, where S. bovinus was the most abundant species (Mahmood et al., 2024)."*

    d. Line 601: Mention that this is hypothesis 2 (here, instead of in the next sentence that only references previous studies).

We changed the text accordingly (See lines 1175-1183)

*"In addition to significant expansions in the MgtE gene family, S. variegatus also possesses significant expansions in the nucleobase:cation symporter-1 gene family (TCDB ID 2.A.39) which is associated with Na uptake. In pure culture, S. variegatus exhibited significantly higher mycelial Ca and Fe concentrations in the gabbro compared to the limited treatment, further supporting our second hypothesis about mineral weathering resulting in base cation uptake when fungi are growing with mineral additions. Previous studies have shown that S. variegatus can take up K+ from granite particles (Fahad et al., 2016), produce high levels of organic acids in the presence of minerals (Olsson and Wallander, 1998; Wallander and Wickman, 1999), and exbibit increased weathering activity when in symbiosis with Pinus sylvestris compared to non-mycorrhizal plants (Wallander and Wickman, 1999). "*

    e. Line 612: Mention that this is hypothesis 3, since hypotheses were numbered in the abstract.

Done (see line 607)

    f. Lines 628, 632: Add references to figures here.

Done (see line 589)

    g. Line 624: Mention that this is hypothesis 4

Hypothesis 4 is removed and added in together with hypothesis 1 as suggested, please see comment above.

    h. Lines 625-634 belong in the results, not the discussion.

This has been moved to the results. See lines 432-435

    i. Line 634: typo?

We have deleted the extra "t"

    j. Lines 649-652: run-on sentence

We have updated the text to (See lines 627-630):

*"Contrary to our hypotheses which were confined to Suillus, we also observed significant expansions and mycelial uptake of elements in the genus Piloderma. Therefore, future controlled pure culture experiments should investigate mineral weathering capabilities across a wider range of species. Additionally, future work on the transcriptional regulation of transporter proteins in Suillus and other ECM species could provide deeper insights into the mechanisms underlying mineral weathering and base cation uptake."*

---

## Author Comment (AC2)

Reviewer 2:

General suggestions:

1. Please add at the beginning of the introduction some information about mineral weathering and perhaps a definition.

We have added a definition of biological mineral weathering at the beginning of the introduction (See line 39):

*"In boreal forests biological mineral weathering, the disintegration and dissolution of minerals caused by biota, is driven by plant-associated ectomycorrhizal (ECM) fungi forming the dominant type of symbiosis primarily with trees of the family Pinaceae (Smith and Read, 2008)."*

Additionally, we have expanded on the mechanisms ECM fungi can use to weather minerals on lines 45-48:

*"To weather minerals, ECM fungi must colonise mineral rich soils and utilise plant allocated C in the application of physical force, the production and exudation of low molecular weight organic acids, free radicals, protons and siderophores, which can break down minerals leading to the mobilisation of base cations (Finlay et al., 2020; Fomina et al., 2010; Schmalenberger et al., 2015)."*

2. I'm wondering if it would be helpful to have a diagram of the soil and the movement of the nutrients through weathering and fungal uptake in the introduction. Perhaps including Mg and the elements in Table 1 to bring the different parts of the paper together that feel a bit dispersive.

Thank you for the suggestion. We have added a figure to aid in the explanation of nutrient mobilisation and uptake during weathering, please see Fig. 1.

3. It is unclear overall what is the logic of the questions. For example, why did the authors look into the fungal mating-type pheromone? Justify the biology? Explain why is the expansion in pheromone receptors suggestive of adaptation to an environment?

We see the confusion of presenting these results unintroduced. We lifted this transporter gene family in our results as it was prominent in our CAFE5 analyses and is involved in sexual reproduction which can influence rates of evolution and adaptation, by increasing the occurrence of genetic recombination in a population. This may in turn have an influence on the development and continuity of mineral weathering capabilities of taxa.

In the introduction to the bioinformatics section in the materials and methods (section 2.1) we highlight that we look into transporter families which are identified as interesting in more detail (see lines 137-138):

*"Additionally, transporter gene families identified as particularly relevant in our analyses were examined in greater detail."*

Furthermore, we have added the following to section 3.1.3.2 (see lines 392-396):

*"In addition to base cation transporter gene families, we also examined transporter gene families that were significantly expanding and/or contracting in our CAFE5 analyses and are associated with processes that may influence evolutionary rates and adaptation, such as those involved in sexual reproduction. Given that shifts in rates of evolution may, in turn, impact the mineral weathering capabilities of taxa, these gene families were considered to be of particular interest.*

*Our CAFE5 results indicated numerous significant expansions and contractions in the fungal mating-type pheromone receptor (MAT-PR) gene family (TCDB ID 9.B.45), which is involved in mate recognition and compatibility."*

*The discussion has also been expanded to properly highlight the role of sexual reproduction in environmental adaptation (see lines 525-533):*

*"Significant expansions of the fungal mating-type pheromone receptor (MAT-PR) gene family were observed in several Suillus species, in both the full and partial family datasets (Fig. S3 and Fig. 3, respectively), with a notable expansion also seen in Russula emetica in the full dataset. This gene family encodes pheromone receptors located on the plasma membrane that play key roles in mate recognition, compatibility, and the reciprocal migration of nuclei during mating (Coelho et al., 2017; Di Segni et al., 2011; James, 2015; Xue et al., 2008). Sexual reproduction, which these receptors facilitate, promotes genetic recombination, enhances the selection of advantageous traits, and accelerates adaptation compared to asexual reproduction (Becks and Agrawal, 2012; McDonald et al., 2016; Nieuwenhuis and James, 2016). It is especially advantageous in unstable or nutrient-variable environments (Nieuwenhuis and James, 2016). Thus, the observed gene family expansions may reflect an increased reliance on sexual reproduction, potentially boosting adaptability in changing environments."*

4. For hypothesis 2, does the base cation uptake simulated in experimental conditions by the mineral additions?

In treatments with mineral additions, the only source of base cations is from the minerals, therefore fungi must weather the minerals to be able to take up any base cations. Then, yes, the base cation uptake is simulated by the mineral additions.

5. Are hypotheses 3 and 4 fundamentally the same? Or is there something specific about Suillus that the authors are trying to test?

Hypothesis 3 *"base cation uptake by Suillus growing in pure culture with mineral additions will be greater compared to other fungal species" is testing whether or not Suillus species are better at taking up base cations from mineral sources than other fungi"* However, the hypothesis does not say anything about the mechanism of uptake.

Hypothesis 4 *"base cation uptake will correlate with base cation transporter gene family copy numbers"* tells us if the mechanistic explanation behind increased mineral uptake is due to increased copy numbers of the transporter families involved.

However, based on a suggestion by reviewer 1, we have now combined hypothesis 1 and 4 and the hypotheses read as follows: (see lines 118-123)

*1) greater base cation uptake is dependent on base cation transporter gene family copy numbers that result from evolutionary expansions, 2) mineral weathering results in base cation uptake by Suillus growing in pure culture with mineral additions, and 3) base cation uptake by members of the genus Suillus, frequently reported to perform mineral weathering, will be greater when growing in pure culture with mineral additions compared to other fungal species. A correlation between base cation uptake and base cation transporter gene family copy numbers would provide a mechanistic explanation for mineral weathering.*

6. If it's just Suillus, how can we know if it's just lineage-specific? How can you exclude the evolution at the root? Or explain more clearly what the analysis is showing within Suillus.

We have expanded our explanation about the CAFE5 analysis and the use of the Agaricomycotina tree. The CAFE5 analysis in the Agaricomycotina tree allows us to compare Suillus with other lineages within Agaricomycotina, which in turn allows to understand whether the changes in copy number are occurring only or mainly in Suillus or not (see lines 176-179):

*"The same data was used to estimate gene family evolution of transporters across the Agaricomycotina phylogeny using CAFE5 v5.1 (Mendes et al., 2020). This approach facilitated the comparison of changes in transporter gene family copy number between multiple lineages, enabling evaluation of whether these changes were isolated to the Suillus lineage or more broadly distributed."*

7. Add the goal/context for the bioinformatics section (2.1) to understand the questions those methods will help address.

We have added a paragraph introducing the bioinformatics used and how they aid in answering our questions in section 2.1 (see lines 132-138):

*"Bioinformatic analyses were performed to investigate whether evolutionary expansions have occurred in base cation transporter gene families in mineral-weathering species, such as those in the genus Suillus, and to assess the relationship between gene copy number and mineral weathering capacity. To achieve this, a phylogeny of Agaricomycotina taxa with diverse lifestyles was constructed. Evolutionary expansions and contractions of all transporter gene families, including those involved in base cation transport, were then analysed to enable a comparative assessment of the copy number variations across the phylogeny. Furthermore, transporter gene families identified as particularly relevant in our analyses were examined in greater detail."*

8. Provide more context for looking into Mg transporter family.

The importance of Mg and studies reporting it's uptake by ECM fungi have been raised in the introduction (see lines 67-69 and 89-93):

*"Mahmood et al., (2024) also showed that mobilisation and uptake of magnesium (Mg), which acts as an activator for many enzymes and plays a key role in photosynthesis, occurred primarily in the deeper mineral horizon and was driven by carbon allocation to ECM mycelium (Black et al., 2007; Bose et al., 2011)."*

*"Suillus species have been observed to increase production and exudation of LMWOAs (Adeleke et al., 2012; Olsson and Wallander, 1998; Wallander and Wickman, 1999), and take up base cations such as potassium (K), Mg and iron (Fe) (Balogh-Brunstad et al., 2008a; Fahad et al., 2016). Mahmood et al., (2024) found that $Mg^{2+}$ was taken up from B horizon soil solution in reconstructed podzol microcosms containing Pinus sylvestris seedlings growing in natural forest soils, where S. bovinus was the most abundant taxon."*

We have also included more context about why Mg and the Mg transporter gene family is of interest in sections 2.1.4 and 3.1.4.

2.1.4 (see lines 208-209):

*"Our CAFE5 results and subsequent correlations between base cations and base cation transporter gene family copy numbers led us to investigate the Mg transporting MgtE transporter gene family further."*

3.1.4 (see lines 411-415):

*"The Mg transporting MgtE transporter gene family was highlighted as particularly relevant by our CAFE5 results, as one of the most frequently significantly expanding and contracting base cation transporter gene families in both the full and the partial datasets. It was also found to be frequently significantly expanding and contracting in the clade containing Suillus species in the partial dataset. Additionally, copy numbers of the MgtE transporter gene family were found to have significant correlations with mycelial Mg concentration in isolates grown in pure culture in axenic systems. Furthermore, Mg plays an important role in plants and fungi, and has been found to be taken up by ECM fungi from minerals and mineral soils (Fahad et al., 2016; Mahmood et al., 2024)."*

9. Why is there no figure with the whole Agaricomycotina tree?

We present a tree of the whole Agaricomycotina in Figure S1. We think that it is not possible to add this tree as a main figure in a way that it fits an A4/letter size paper and that it can be easily read and interpreted. Therefore, we have decided to keep it as a supplementary information so that the reader can have access to it and can zoom in to relevant parts.

10. I have a few specific suggestions:

a. Figure 1 - what is the input matrix to calculate the PCAs? counts of genes? Perhaps a multipanel figure with also the phylogeny might help explain the analysis?

We have added in the figure legend (see Fig. 3)(lines 385-390) and the text (lines 268-172) that copy number counts per transporter family were used as input, also we have included a table in the supplement with the copy number count data per transporter gene family per species (Tab. S3).

*"Relationships between transporter gene family copy numbers and fungal taxa were analysed and visualised by performing a principal component analysis (PCA) on count data of transporter gene family copy numbers (Tab. S3) using CANOCO 5.10 (Microcomputer Power, Ithaca, NY, USA; Braak and Smilauer, 2012)."*

b. Figure 2 - add branch lengths. The reason for this tree is to show the taxa that are tested in the lab? Since the bioinformatics section talks about a Agaricomycotina tree, I am puzzled by the selection of these taxa and what the purpose of this tree is.

We have presented a tree of the whole Agaricomycotina (the full dataset) with true branch lengths in figure S1. The reason for not including it as a main figure is because some of the branch lengths (particularly in the Suillus clade) are very short and it is difficult to see the branches properly.

We have clarified that both a full dataset with all 108 Agaricomycotina taxa and scaled copy number values, and a partial dataset with 28 taxa and true count values of copy numbers were used and their usefulness. We have also expanded on our explanation for the selection of the 28 taxa used in the partial dataset (see lines 184-193):

*"Although the values are scaled, it remains valuable for observing expansions and contractions across the entire Agaricomycotina phylogeny.*

*"CAFE5 was also run on a smaller dataset with true count values of copy numbers to account for any error introduced by the scaling and allow for observation of expansions and contractions calculated with true copy number values in key species of interest. Twenty-eight taxa were selected for this smaller dataset including all Suillus species, as they have been frequently reported to perform mineral weathering and have been found to inhabit the mineral B horizon in boreal forest soil profiles. Piloderma species were selected as they are also ECM fungi and have been found to inhabit both the organic and mineral horizons in boreal forests. Finally, two saprotrophic fungi, Coniophora puteana and Serpula lacrymans, were selected to allow comparison between the ECM and saprotrophic lifestyle."*

Additionally, we have elaborated on the reason for the selection of these species in section 2.2 (see lines 218-223):

*"The ECM genus Suillus has been reported to possess mineral weathering capabilities and inhabit the mineral horizon in boreal forest soils, and was therefore studied in greater depth. Isolates of Suillus, the ECM genus Piloderma, which inhabits both the organic and mineral*

*soil horizons (Mahmood et al., 2024; Marupakula et al., 2021), and two saprotrophic fungi,
Coniophora puteana and Serpula lacrymans, were included to compare between ECM and
saprotrophic lifestyles. All isolates were grown in pure culture axenic systems with and
without mineral additions to determine base cation uptake during mineral weathering."*

      c. Figure 3 - highlight panels with significant slopes and mention statistical test
          in caption.

We have added the statistical test used and highlighted significant panels in the figure
legend (Now Figure 4; lines 434-436):

*"Pearson's correlations between mycelial magnesium (Mg), calcium (Ca) and iron (Fe)
concentrations (mg/kg) and transporter gene copy numbers. Element concentration is given
as the estimated mean values in eight different fungal species grown in pure culture in
gabbro, granite, limited and rich treatments, and the gene copy numbers of the Mg2+
transporter-E family (TCDB ID: 1.A.26) in the corresponding fungal genomes. Significant
correlations with p < 0.05 are found in panels a, b, c and g."*

Additionally, we realised that we miswrote in the materials and methods section where we
said that a Spearman's correlation was used, but it was in fact a Pearson's (see lines 304-
305):

*"Correlations between base cation transporter gene family copy numbers and means of
mycelial elemental concentration for all isolates in each mineral treatment and the control
treatments were tested using Pearson's correlation."*

      d. Figure 4 - highlight different species on the figure with boxes around the
          names or labels.  Report significance and statistical tests used in the caption.

Figure 4 (now figure 6) has been adjusted to make it clearer which species different isolates
belong to and with the addition of a horizontal line to highlight isolates which are
significantly different from each other.

The materials and methods have been adjusted to describe more explicitly how the ratios
were estimated and reported (see lines 300-303):

*"Ratios were calculated from the means of all replicates per isolate per treatment, yielding
one ratio per comparison. Ratio means are reported with 95% confidence intervals, based on
model uncertainty and pooled variance across isolates. Ratios with non-overlapping
confidence intervals were considered significantly different."*

The figure legend has also been adapted (see lines 460-468):

*"Figure 5 – Ratios of mycelial calcium (Ca) concentration (mg/kg) between gabbro and
limited treatments for each fungal isolate. Isolates include Suillus bovinus (Sb1–4), S. luteus
(Sl1–3), S. variegatus (Sv1–3), S. granulatus (Sg1–3), S. grevillei (Sg4), Piloderma byssinum
(Pb), P. aff. fallax (Pf), P. olivaceum (Po), P. sphaerosporum (Ps), Piloderma
sp. 1, Piloderma
sp. 2, Coniophora puteana (Cp), and Serpula lacrymans (Sl). Data points represent the mean*

*ratio of Ca concentration between gabbro and limited treatments for each isolate, with error bars indicating the 95% confidence interval. Isolates with non-overlapping confidence intervals are considered significantly different from one another. The red dashed line represents a ratio of 1.78, distinguishing isolates with significantly higher Ca concentrations above the line (S. luteus, S. variegatus, and all S. bovinus isolates except Sb16) from those below it, which include all Piloderma species except P. sphaerosporum."*

e.  L18 list species of saprotrophic fungi

We have now added "*Coniophora puteana*" and "*Serpula lacrymans*" to line 18.

f.  L397-401 sentence is incredibly long

We have re-written the long sentence (see lines 423-425):

*"Of the 25 base cation transporter gene families that were found to be significantly expanding or contracting, 11 showed significant correlations between gene family copy number and base cation uptake. From these, six are specifically involved in the transport of Ca, Fe, K, Mg, Na and P and five are base cation transporters not linked to a specific base cation. "*

L417 onwards, is there a way of summarising the information on significance so that it's not so difficult to read through? perhaps a table?

We have included a table (Tab. 2) summarising the R values with an asterisk indicating a significance of $p \leq 0.05$ to make it easier to read (see lines 440-445)

*"Table 2 – Summary of R values for each of the correlations between estimated mean mycelial elemental concentration and base cation transporter gene family copy numbers of eight different fungal species grown in pure culture in gabbro, granite, limited and rich treatments. Element, Treatment and TCDB IDs of base cation transporter gene families are indicated. R values with significant in bold with p values where $p \leq 0.05$ indicated with an asterisk. Correlations were not performed for cases where a base cation transporter gene family is not associated with the transport of the respective element, as indicated by empty cells."*

g.  L447 italicise luteus

We have now italicised *luteus*.

---

## Author Response (AR2)

Reviewer 1:

Overall, this revised version reads considerably better, particularly in the introduction, discussion, and updated figures. I have a few (mostly minor) suggestions that I believe should be addressed before publication.

We thank the reviewer for their valuable comments. We have carefully considered their suggestions and, in agreement with the other reviewer's feedback, have made alterations throughout the manuscript to improve clarity and reduce its length. Please see below for specific replays.

Abstract
1. In my opinion, 475 words is way too long for an abstract. Try reducing it to around 300 words, highlighting only the major takeaways from this study.
   Yes, we agree with you that the abstract is much too long. We have now reduced it to 300 words, highlighting only the key parts of our study (lines 10-28).

2. Lines 19-20: Put the results after the hypotheses.
   In the shortened version of the abstract, the techniques used are before the hypotheses and the results are after.

Intro
1. Lines 57-59: Awkward wording.
   In our efforts to reduce the volume of text (in accordance with reviewer 2) this sentence has been removed.

2. Overall, the additions/changes (including the addition of Figure 1) provide a clearer background for the reader.
   We appreciate the comment and are glad that our changes improved the readability of the manuscript.

Methods
1. Please describe the significance threshold used for determining a significant expansion or contraction in copy number for a gene family.
   We have now included the following sentence to describe the significance threshold and to clarify that significance is assigned to rate of evolution which is inferred from the numbers of contractions and expansions (lines 152-153):

   *"Rapidly evolving gene families were identified as those that have a significantly (p < 0.05) higher rate of evolution than the mean rate. This rate of evolution is inferred from the numbers of expansions (gains) and contractions (losses)."*

2. For most of the bioinformatics analyses, it is not specified whether DNA or amino acids were used. Please clarify these.
   We have now added that amino acid sequences were used to identify orthogroups and reconstruct the phylogenetic relationships of Agaricomycotina (lines 116-117):

   *"Orthogroups were identified from amino acid sequences with OrthoMCL v2.0.9"*

And (lines 180-183):

*"To explore the evolution of this gene family, a phylogenetic tree of all MgtE family amino acid sequences present in the 28 taxa of the partial family dataset was constructed."*

Results and figures
1. Fig. S1: Not every branch has a bootstrap value. For these, was no value assigned, or was it not significant enough? A minor detail, but worth clarifying briefly.
   We have repositioned the bootstrap values so they are more clearly associated with each branch. They now sit above each branch or next to it indicated with a line.

2. Paragraph beginning at line 350: There are several mentions of "most frequent transporter gene families". I assume these are the gene families that gained/lost the most copies? Please clarify.
   Here what we mean is that these families had the highest number of significant gene gain or loss events. We have adjusted the text to make this clearer (lines 304-306):

   *"Across the whole phylogeny, the MgtE gene family possessed the largest number of expansion or contraction events, alongside the MSF gene superfamily in the full dataset, and the β-amyloid cleaving enzyme 1gene family (TCDB ID 1.A.33) in the partial dataset."*

3. Add 2+ or + superscripts to Mg and other cations, they are omitted in several places.
   Thank you for pointing this out. We have added superscripts to all instances where we are specifically talking about ions and not elements.

4. Table 2 legend: Pearson correlation coefficient (r) should be lowercase to distinguish it from R2 used in a linear regression.
   Thank you for pointing that out, it is now a lowercase "r".

Discussion
1. The interpretation of MAT-PR expansions in several Suillus sp enhancing sexual reproduction makes sense and is a great addition to the discussion.
   We appreciate the comment and we are pleased that this is seen as a valuable addition.

2. More genes, more cation uptake- this makes sense. However, is there evidence for specific variants of certain genes leading to more efficient uptake? Possibly an interesting point to add.
   We weren't able to experimentally determine this here with our genes of interest but we have added that the sequence similarity for MgtE genes is fairly low and that this may indicate that they are not functionally redundant (lines 477-479):

   *"Furthermore, when comparing within each taxon, MgtE gene amino acid sequences generally had low similarity, with the majority having around 30-40 % shared identity. This may indicate that the genes of the MgtE family are, like those of the MIT gene family in plants, not functionally redundant."*

3. Line 535: typo

We didn't find a typo on this line, however we did find one on a nearby line "nutirental-variable" (lines 458-459):

*"It is especially advantageous in unstable or nutritionally variable environments"*

Also for the figure reference "S57" which has been changed to "S5" (line 477), and we corrected the spelling for the name of one of the taxa "*S. cf subluteus*" to "*S. cf. subluteus*" (line 467).

4. Lines 594-599: this sentence should be split up
   Thank you, we have now split up this sentence to improve the readability (lines 517-519):

   *"For example, the oligopeptide transporter (OPT) family, which is associated with Fe transport (Kurt, 2021; Mousavi et al., 2020; Wintz et al., 2003), was significantly negatively correlated with Mg uptake in the gabbro, granite and limited treatments. This may indicate an inhibitory effect of proteins of this family on Mg uptake (data not shown)."*

5. Lines 617-617: This last sentence leaves me a bit confused. Don't the results of figure 4 suggest that increased copy numbers of MgtE correlate with increased cation uptake in Piloderma and other species as well?

   Yes, we have now adjusted the text in the conclusion to be clearer and better present our interpretations (lines 537-551):

   *"...Base cation transporter gene families were among the most abundant transporter types, showing rapid evolution not only in Suillus, but also in the genus Piloderma. The MgtE gene family, in particular, exhibited numerous expansions and contractions across the whole phylogeny. Mycelial uptake of Mg in the presence of gabbro and granite, increased sharply with copy numbers of MgtE genes, and significant correlations were found in the gabbro, granite and limited treatments. Our findings highlight potential associations rather than definitive functional assignments, indicating that while gene copy number correlates with base cation uptake capacity, causal relationships remain to be validated experimentally.*

   *Although this study focused primarily on Suillus, we found that Piloderma species also displayed significant expansions in transporter gene families, showed comparably similar base cation uptake to Suillus in many instances, and strongly contributed to the positive correlations observed between base cation uptake and base cation transporter gene family copy number. Future studies should examine mineral weathering capabilities of Piloderma species alongside Suillus species, and controlled pure culture experiments should incorporate a broader range of ECM fungi to assess the prevalence of mineral weathering in this guild. Additionally, investigating the transcriptional regulation of transporter proteins in Suillus and other ECM species could provide deeper insights into the mechanisms driving mineral weathering and base cation uptake."*

Reviewer 2:

In their work, King et al. investigated mineral weathering by ECM fungi from both an evolutionary and an experimental perspective. They conducted a bioinformatic analysis focusing on expanding and contracting genes families, with particular attention to on base cation transporter gene families and the genera Suillus and Piloderma. They also grew several fungal species in a setup which allowed them to observe and quantify weathering capabilities of each species. Their final aim was to combine the bioinformatic analyses with growth assays, thereby testing for a direct connection between base cation uptake transporters and weathering.

I found the paper somewhat difficult to read and understand. I am not a specialist in phylogeny or bioinformatics, and ECM-related mineral weathering is also outside my main expertise. Still, I think some of the difficulty comes from the way the aims and outcomes are described. On close reading, it seemed to me that the authors could not directly link individual transporter genes to function. This is absolutely fine – negative or inconclusive results are part of the scientific process – but I think it should be stated more openly. Otherwise, the reader invests a lot of effort and may in the end feel a bit "tricked".

Beyond that, the manuscript feels rather long, and several sections cold be condensed, as they seem to stray from the main topic. I provide more concrete suggestions in the specific comments below.

We thank the reviewer for their thoughtful comments and for taking the time to evaluate our manuscript, even though this topic lies outside their primary area of expertise. We appreciate the reviewer's observation that parts of the text could be made clearer to non-specialists, and we have revised the different sections to better communicate the study's aims and outcomes.

Regarding the point that individual transporter genes could not be directly linked to specific functions, we agree with this assessment and have now stated it more explicitly in the revised manuscript (for example see lines 541-543). We have clarified that our findings highlight potential associations rather than definitive functional assignments, and we emphasize that such results are an important step toward hypothesis generation for future experimental work.

*"Our findings highlight potential associations rather than definitive functional assignments, indicating that while gene copy number correlates with base cation uptake capacity, causal relationships remain to be validated experimentally."*

Finally, we have shortened the text (by 2000 words from the original submission) to enhance focus and readability. We believe these changes have improved the manuscript's clarity, flow, and overall coherence.

Specific comments and suggestions:

1. 18: Since the fungi were always grown in pure culture, it may not be necessary to repeat "growing in pure culture" every time.

   We appreciate the reviewer's suggestion. However, we prefer to retain the phrase "growing in pure culture" throughout the manuscript to emphasize that all fungal experiments were conducted under controlled, axenic conditions. This clarification is

important for distinguishing our results from others that can be obtained in symbiotic or environmental contexts. In addition, shortening the text down has also taken some of these formulations away.

2. 60 – 75: This section could be shortened. It describes the effects of C and N availability on weathering, which doesn't seem like the main focus of your experiment.
Thanks for the suggestions. We have now shortened this section.

3. 73 – 75: It's unclear what exactly needs to be understood better. If you specify – for example, it should be temporal variations, interspecific or intraspecific differences, somethis else… - the paragraph would have a clearer conclusion.
We have now modified the text to make what needs to be understood clearer (lines 46-48):

*"it is therefore essential to improve our understanding of ECM-mediated mineral weathering and base cation uptake, including which ECM species play key roles and how these processes are regulated to aid the development of more sustainable forestry practices."*

4. 84: The extra sentence about LMWOA may not be necessary, as this is not the focus of your experiments.
In our efforts to reduce the volume of text this sentence has been removed.

5. 94: Mentioning the two additional mineral-weathering species seems unnecessary since they are not included later in analyses or discussion.
We have chosen to include these two species to highlight that it is not only *Suillus* species that perform mineral weathering. We have also added the *Piloderma* genus to the list (lines 53-57):

"Several ECM species have been identified to perform mineral weathering including *Paxillus involutus* (Bonneville et al., 2009, 2016; Gazze et al., 2012; van Scholl et al., 2006), *Pisolithus tinctorius* (Balogh-Brunstad et al., 2008b; Paris et al., 1995, 1996), a number of *Piloderma* species (Glowa et al., 2003; van Scholl et al., 2006), as well as a number of *Suillus* species which are commonly found in the B horizon in boreal forests (Lofgren et al., 2024; Mahmood et al., 2024; Marupakula et al., 2021)."

6. 100: Suggestion for clarity and flow: "can take place at multiple stages, from genomic to post-translational. On the genomic level, one mechanism is …"
Thank you for your suggestion, we have now incorporated it into the text (lines 71-76):

*"Regulation of base cation uptake through transporter proteins can take place at multiple stages, from the genomic to post-translational stage (Mukhopadhyay et al., 2024)... ...On the genomic level, one mechanism of regulating transporter protein expression is differences in the number of copies of a protein encoding gene."*

7. 101: Here CNV is mentioned for the first time. A short clarification would help: (for example "CNV refers to differences in gene copy number within a species (among

individuals or populations). This does not equal long-term gene family expansion, which is the evolutionary outcome of repeated duplication events"). Adding this distinction would make the text easier to follow.

We have modified the text to shift the focus to differences in copy number between species and highlight that CNV is within species (lines 75-78):

*"On the genomic level, one mechanism of regulating transporter protein expression is differences in the number of copies of a protein encoding gene. This can occur within species or populations and has been shown to drive genome evolution and environmental adaptation (Steenwyk and Rokas, 2017; Tralamazza et al., 2024). Differences in copy number can also occur between species, and Wapinski et al. (2007) found that genes..."*

8. 111 – 115: The sentences here are unclear. Is $K^+$ transporter regulation the only case identified so far? For the second sentence, a shorter version would improve flow: "Yet, no studies have addressed the evolution of these transporter genes in the context of ECM-mediated mineral weathering."

As far as we know, $K^+$ transporter genes are the only base cation transporters currently identified which have been upregulated during mineral weathering. We have moved the first sentence up in the text to improve the flow of the paragraph (lines 71-75):

*"... from the genomic to the post-translational stage (Mukhopadhyay et al., 2024). Although studies on the regulation of base cation transporter genes are limited, transcriptomic analyses have revealed that $K^+$ transporter genes are upregulated during mineral weathering in Hebeloma cylindrosporum (Corratgé et al., 2007; Garcia et al., 2014), Amanita pantherina (Sun et al., 2019) and Paxillus involutus (Pinzari et al., 2022). On the genomic level, one ..."*

We have also adjusted the second sentence to improve the readability (lines 86-88):

*"Currently however, there are no studies addressing the evolution of base cation transporter genes involved in ECM-mediated mineral weathering or how they might contribute to base cation uptake regulation."*

9. 120: Again, repetitions of "pure culture" can be avoided.

Please see our replay above. Here we state pure culture in subheadings and in figure legends to make clear what system we are using, thinking that some reader that scan the paper will not be led to think that it is done in symbiosis.

Materials & Methods:

10. 163: What are the three levels of criteria? If they matter, please list them briefly; if not, the remark can be removed.

This part has been removed to avoid confusion.

11. 164: Numbers (e.g. 173) are clearer than words ("one hundred and seventy-three").

In our efforts to reduce the amount of text this part has been removed.

12. 170: Suggested rewrite to avoid repetition: "Relationships between transporter gene family copy numbers and fungal taxa were analysed and visualised using principal component analysis (PCA) on count data (Tab. S3) with CANOCO 5.10 (Microcomputer Power, Ithaca, NY, USA; Braak and Smilauer, 2012)."
We have now modified the sentence to avoid repetition (lines 142-144):

*"Relationships between transporter gene family copy numbers and fungal taxa were analysed and visualised using principal component analysis (PCA) on count data (Tab. S3) using CANOCO 5.10 (Microcomputer Power, Ithaca, NY, USA; Braak and Smilauer, 2012)."*

13. 179: A short explanation of "birth" and "death" in CAFE5 would help (less knowledgeable) readers: e.g. "Due to the birth–death model (duplication rate – loss rate) employed by CAFE5…"
We have now added the following sentences (lines 152-153):
*"Rapidly evolving gene families were identified as those that have a significant (p < 0.05) higher rate of evolution than the mean rate. This rate of evolution is inferred from the numbers of expansions (gains) and contractions (losses)."*

This is in response to a comment from reviewer 1 about indicating the significance value, but it also includes an explanation of what we mean by rapidly evolving and how this is inferred, which we agree that will help non expert readers to understand the cafe analysis.

14. 199: Please clarify what the four levels of criteria are, or if they don't matter, the remark can be removed.
This has now been removed to avoid confusion.

15. 200: A schematic of the three datasets, showing how they relate to each other and how they are used in analyses, would greatly help the reader.
We have opted to simplify and improve our explanation of the three datasets used (lines 155-171):

*"CAFE5 was unable to run when within-family size variation exceeded 80 copy numbers, so transporter gene family copy numbers were scaled to values of 0-80 and rounded up to the nearest whole number in R v4.3.2 (R Core Team, 2023), using the 'Tidyverse' suite v2.0.0 (Wickham et al., 2019) and scales (Wickham et al., 2024). This dataset is referred to as the "full" dataset and although the values are scaled, it remains valuable for observing expansions and contractions across the entire Agaricomycotina phylogeny.*

*CAFE5 was also run with true count values of copy numbers and a reduced phylogeny of 28 taxa to account for any error introduced by the scaling, and to allow for the observation of expansions and contractions calculated with true copy number values in key species of interest. This dataset is referred to as the "partial family" dataset and included 126 transporter gene families in the CAFE5 analyses. The 28 taxa selected included all Suillus species – reported to perform mineral weathering and to inhabit the mineral B horizon in boreal forest soils, all Piloderma species – found to inhabit both the organic and mineral horizons in boreal forests, and two*

*saprotrophic fungi,* Coniophora puteana *and* Serpula lacrymans – *to allow comparison between the ECM and saprotrophic lifestyle. These taxa were extracted from the original Agaricomycotina tree using R and the packages Geiger (Harmon et al., 2008) and ape v5.8 (Paradis and Schliep, 2019). Copy numbers of subfamilies of significantly expanding and/or contracting transporter gene superfamilies in the partial family dataset were also run with the reduced phylogeny of 28 taxa in CAFE5. This dataset is referred to as the "partial subfamily" dataset, with 51 transporter gene families included in CAFE5 analyses (Tab. S4)."*

We hope these changes are satisfactory.

16. 219: Suggested edit: "Isolates of Suillus and another ECM genus, Piloderma, which inhabits both organic and mineral soil horizons, as well as two saprotrophic fungi, Coniophora puteana and Serpula lacrymans, were included…" (to avoid suggesting Suillus is not ECM).
We have adjusted the text to make sure it is clear that *Suillus* is an ECM genus (lines 188-189):

*"Isolates of* Suillus*, and another ECM genus* Piloderma*, which inhabits both the organic and mineral soil horizons"*

17. 242: Possible typo: should the fraction size be 63–2500 μm (sand particle size)? Also, since you later call it "sand," clarify this earlier: "to retain a sand-sized fraction (63–2500 μm) to maximise surface area…"
This was not a typo, but we see the issue that this size fraction should not be referred to as "sand". We have changed all instances of "sand" to "rock particles".

Results:

18. 351-372: These paragraphs are difficult to follow due to switching between the main dataset and subsets. It may be clearer to structure them by transporter gene family, describing and summarizing trends for each.
We have considerably reduced and restructured this text in an attempt to improve the flow. While we have maintained the main structure of discussing the different datasets, as this approach allows us to identify common trends that are summarized later in the manuscript, we have clarified and condensed the sections to make them easier to follow.

19. 418: Suggested edit for clarity: "The phylogeny showed the latter: several clades of MgtE genes are present in each taxon."
Thank you for the suggestion, we have now altered the text (lines 344-345):

*"The results supported the latter, revealing several distinct clades of MgtE genes present in each taxon"*

20. 423: Suggested rewrite: "Of the 25 base cation transporter gene families that were significantly expanding or contracting, 11 also had copy numbers significantly correlated with base cation uptake."

We have rewritten this sentence, however we have elaborated to make it clear that correlations were drawn between base cation transporter gene family copy numbers and base cation concentrations for specific elements (lines 349-351):

*"Of the 25 base cation transporter gene families that were significantly expanding or contracting across all three datasets, 12 showed significant correlations between mycelial base cation concentrations and transporter gene families that transported the corresponding base cation."*

21. 426: The relevance of reporting all 28 significant correlations is not clear. Many (16/28) correlations are under control conditions (which don't directly prove weathering), several are negative, and for some TCDB IDs it is unclear how they relate to the reported ion transport. See also my comments on Table 2.

We see that it would be more useful to report correlations in the mineral treatments, and we have amended the sentence (lines 352-354):

*"Across all treatments there were 28 significant correlations, of which 23 were positive. Of these, nine were in either the gabbro or the granite treatment."*

22. 443: Questions regarding Table 2 (as I was not able to find a confirmation of these):
    a. • Are 2.A.3 or 2.A.39 involved in Ca transport?
    b. • Is 3.A.1 involved in Mg and Na transport?
    c. • Are 3.A.3 or 3.A.3.2 involved in Mg transport

We have aimed to clarify this by improving our explanation of the correlations that were made in the materials and methods (lines 265-269):

*"Analyses were restricted to base cation transporter gene families which were both identified as significantly expanding or contracting in allCAFE5 analyses, and associated with the uptake of $Ca^{2+}$, $Fe^{2+}$ or $Fe^{3+}$, $K^+$, $Mg^{2+}$, $Na^+$, or P. This was determined using the TCDB substrate search tool (https://tcdb.org/progs/?tool=substrate#/ ). Correlations were only performed between mycelial base cation concentrations and transporter gene families that transported the corresponding base cation."*

We have also adjusted the table legend to be clearer (lines 366-369):

*"Base cation transporter gene families were only correlated with the elements they are associated with the uptake of. The table indicates element, treatment and TCDB IDs (see Tab. S2 and S4) of the transporter gene families. R values with statistically significant p values ($p \leq 0.05$) are indicated in bold and with an asterisk. Empty cells denote that the transporter gene family in that column does not transport the element listed in the corresponding row"*

23. 463: Isolate IDs are mislabeled (for example, S. bovinus are labeled as Sb4, Sb16, Sb25 and Sb36 but described as Sb1-Sb4 in the caption; same goes for all other isolate IDs)

Thank you for pointing this out, we have now fixed the figure. We also noticed the same mistake in figures S7 and S8, which we have also corrected.

24. 477: Reporting here doesn't fully match Figure 5. The figure shows significant ratios for 10/14 Suillus and 5/6 Piloderma isolates, but this is not stated. Also, C. puteana is omitted, though its isolate also showed a significant ratio.
    We have modified the text to better highlight that the figure is showing ratios which are significantly different from each other as indicated by non-overlapping confidence intervals. Additionally, *C. putenana* has been included in the text (lines 395-398):

    *"The strongest difference between groups was seen in Ca. Furthermore, for Ca ratios between the gabbro and limited treatments, all S. bovinus, S. luteus and S. variegatus isolates, as well as C. puteana exhibit significantly higher values than all but one Piloderma isolate, as indicated by non-overlapping 95 % confidence intervals in Figure 5"*

    We have also altered the text following this sentence to better reflect the data in figures S7 and S8 (lines 398-401):

    *"For Mg ratios between both mineral treatments and the limited treatment, the isolate S. luteus 2 had a significantly higher ratio between gabbro and limited treatments for Mg and P concentrations, and between granite and limited treatments for Mg, compared to all other isolates (Fig. S7 and S8). In all other cases, for both mineral treatments, isolates showed similar variation in ratios within and between genera (Fig. S7 and S8)."*

25. 479: Results do not mention that C. puteana had a high gabbro:limited Ca ratio. Since this species is part of your saprotrophic comparison, this seems important to highlight.
    We have now highlighted this (lines 396-397):

    *"Furthermore, for Ca ratios between the gabbro and limited treatments, all S. bovinus, S. luteus and S. variegatus isolates, as well as C. puteana exhibit significantly higher values than all but one Piloderma isolate..."*

Discussion:

26. General note on discussion: Overall, many conclusions seem only loosely supported by the results.
    We have modified the discussion throughout to either back up our conclusions more strongly, to propose alternative explanations for our results, or to adjust the strength of our claims. For example see lines (495-500)

    *"In pure culture however, S. variegatus exhibited significantly higher mycelial Ca and Fe concentrations in the gabbro compared to the limited treatment, further supporting our second hypothesis. Like S. bovinus and S. luteus, the observed Ca and Fe uptake could indicate that expansions of Ca and Fe transporting gene families in the ancestors of S. variegatus were already sufficient to support mineral weathering and further expansions were not required. It could also be that another form of regulation, such as transcriptional up-regulation, may play a role in the uptake of Ca and Fe in S. variegatus."*
    And lines (526-533)

*"The comparison of base cations and P ratios in the mineral and limited treatments supports our second hypothesis. However, these comparisons only partially support our third hypothesis that base cation uptake by Suillus species would exceed that of other taxa when grown with mineral additions. A clear pattern emerged only for Ca in ratios gabbro and limited treatments, where Suillus species as a group exhibited significantly higher ratios than most other taxa, supporting our third hypothesis. For other base cations, the ratios between mineral and limited treatments showed either no pattern or mixed results, with several Piloderma isolates performing equally well compared to the Suillus isolates, contradicting our third hypothesis. Together with evidence of rapid evolution in multiple base cation transporter gene families in Piloderma, these findings suggest that enhanced mineral weathering capability may extend beyond Suillus to include Piloderma as well."*

27. 520: The 28 significant correlations are, as far as I can say, not sufficient to support Hypothesis 1 (see earlier comment for line 426).
We see that we are supporting our first hypothesis too quickly here and have adjusted the text the properly reflect what is supported by the data. We have taken away the concluding sentence of this paragraph and have changed the 28 significant correlations to the 23 significant positive correlations and highlight the 9 which were in the mineral treatments. We also highlight that greater copy numbers contributing to greater base cation uptake is only supported in some cases, not all, and offer that an alternative level of regulation may play a larger role in these cases. Finally, we have added to the sentence about the rapid evolution of base cation transporter gene families in *Suillus* and suggest that the evolution of these families is not random and aids in species specific adaptation (lines 437-446):

*"Our analyses of 108 Agaricomycotina genomes confirm significant expansions and contractions in base cation transporter gene families, highlighting their rapid evolution within the genus Suillus and indicating that the evolutionary changes in copy number are not random and are likely related to species specific adaptation. Additionally, we identified 23 significant positive correlations between different elements and the copy numbers of expanding or contracting base cation transporter gene families associated with their uptake, nine of which were in the mineral treatments. These findings suggest that increased copy numbers of certain base cation transporter gene families may contribute to enhanced uptake of specific base cations, however this pattern is not universal. This offers partial support for our first hypothesis that greater base cation uptake is dependent on base cation transporter gene family copy numbers which have resulted from evolutionary expansions. In cases where no significant correlation was found, other levels of regulation such as transcriptional up-regulation may play a larger role, and this requires further investigation"*

28. 561-565: For S. bovinus, no expansions were found in base cation transporter families, yet significantly higher Ca, Mg, and Fe were observed in gabbro versus limited treatment. This appears to contradict Hypothesis 1. The statement that this supports previous findings seems unrelated.
Yes, the lack of expansions in *S. bovinus* is not in line with our first hypothesis. We have now adjusted the text to better reflect that and offer some alternative explanations for the lack of expansions. We have incorporated the following paragraph into this one as the same alternative explanations apply. We have also

removed the last sentence about the uptake seen in *S. bovinus* being in line with previous findings (lines 489-493):

*"The significant uptake of base cations, despite the lack of expansions in base cation transporter gene families in S. bovinus and S. luteus may reflect that strong mineral weathering capability was already present in their ancestral lineage, reducing the necessity for additional expansions. It could also indicate that another level of regulation may play a larger role than gene copy numbers in base cation uptake in these taxa"*

29. 568-577: S. luteus has no significant expansions yet shows significantly higher Ca and Fe uptake in gabbro, which again contradicts Hypothesis 1. However, this is presented as supporting Hypothesis 2. The paragraph goes on by referring to other publication's results and concludes by saying "these findings align with our first hypothesis", which seems inconsistent.
We have combined this paragraph with the previous paragraph (lines 487-493):

*"Similarly, S. luteus, which exhibited only one significant contraction in the full dataset, had significantly higher mycelial concentrations of Ca and Fe in the gabbro treatment compared to the limited treatment. Together, these findings support our second hypothesis that mineral weathering results in base cation uptake by Suillus species when grown with mineral additions. The significant uptake of base cations, despite the lack of expansions in base cation transporter gene families in S. bovinus and S. luteus may reflect that strong mineral weathering capability was already present in their ancestral lineage, reducing the necessity for additional expansions. It could also indicate that another level of regulation may play a larger role than gene copy numbers in base cation uptake in these taxa."*

Also, we have now adjusted and moved the part referring to the *S. luteus* heavy metal publications to the second paragraph of the introduction, as we believe it supports our rationale for investigating copy number expansions (lines 432-437):

*"Transporter genes, which are often involved in abiotic and biotic environmental interactions, are therefore expected to undergo frequent expansions and contractions, enabling rapid adaptation to environmental changes and stressors. For example, Suillus luteus has been shown to possess single nucleotide polymorphisms and gene copy number variation in genes involved in heavy metal homeostasis, which includes metal ion transporter genes, enabling tolerant populations to grow under exposure to heavy metals (Bazzicalupo et al., 2020; Colpaert et al., 2011; Ruytinx et al., 2011, 2017, 2019)."*

30. 579-583: S. variegatus shows expansions in MgtE and Na-related genes but actually takes up Ca and Fe. This also seems inconsistent with Hypothesis 1, yet the paragraph mainly cites other studies.
We have now altered the text to highlight the inconsistency and offer an alternative explanation about the lack of expansions in Ca and Fe transporting families (lines 495-500:

*"In pure culture however, S. variegatus exhibited significantly higher mycelial Ca and Fe concentrations in the gabbro compared to the limited treatment, further*

*supporting our second hypothesis. Like S. bovinus and S. luteus, the observed Ca and Fe uptake could indicate that expansions of Ca and Fe transporting gene families in the ancestors of S. variegatus were already sufficient to support mineral weathering and further expansions were not required. It could also be that another form of regulation, such as transcriptional up-regulation, may play a role in the uptake of Ca and Fe in S. variegatus.”*

31. 586: Mention of S. lakei, S. cothurnatus, and S. cf. subluteus feels disconnected from the rest of the discussion.
These species are mentioned earlier in the discussion and, like *S. variegatus*, possess several expansions in base cation transporter families, therefore we feel they are relevant and we opted for keeping them in the discussion.

32. 594: The phrase "not all transporter gene families associated with a particular base cation showed a significant positive correlation" seems misleading; in fact, only a small minority did.
We have now removed this part of the sentence and adjusted the paragraph to better reflect that the first hypothesis isn't universally supported by data from all gene families, but only some (lines 519-523):

*"Taken together, these findings indicate that gene copy numbers may play a key role in the regulation of base cation uptake by certain base cation transporter gene families, like the MgtE gene family, and offer circumstantial support for the first part of our first hypothesis, that greater base cation uptake is dependent on base cation transporter gene family copy numbers. Regulation of base cation uptake by other gene families may take place at the transcriptional or translational level.”*

33. 607-617: This section is hard to follow. The hypothesis was that Suillus species would show greater base cation uptake than Piloderma. Only Ca showed this pattern, so the claim that Suillus has stronger weathering ability seems based only on Ca. The final sentence ("if we broaden our hypothesis…") comes across as uncertain—almost as if the authors themselves are not sure whether Suillus performs better.
We have now altered the text to not overstate the degree of support our data offers our hypotheses and to improve the clarity of the text (lines 526-533):

*"The comparison of base cations and P ratios in the mineral and limited treatments supports our second hypothesis. However, these comparisons only partially support our third hypothesis that base cation uptake by Suillus species would exceed that of other taxa when grown with mineral additions. A clear pattern emerged only for Ca in the gabbro:limited ratio, where Suillus species as a group exhibited significantly higher ratios than most other taxa, supporting our third hypothesis. For other base cations, the ratios between mineral and limited treatments showed either no pattern or mixed results, with several Piloderma isolates performing equally well compared to the Suillus isolates, contradicting our third hypothesis. Together with evidence of rapid evolution in multiple base cation transporter gene families in Piloderma, these findings suggest that enhanced mineral weathering capability may extend beyond Suillus to include Piloderma as well.”*